# Tensor Product Attention Is All You Need

## Abstract

Scaling language models to handle longer input sequences typically necessitates large key-value (KV) caches, resulting in substantial memory overhead during inference. In this paper, we propose **T**ensor **P**roduct **A**ttention (TPA), a novel attention mechanism that uses tensor decompositions to represent queries, keys, and values compactly, significantly shrinking KV cache size at inference time. By factorizing these representations into contextual low-rank components (contextual factorization) and seamlessly integrating with RoPE, TPA achieves improved model quality alongside memory efficiency. Based on TPA, we introduce the **T**ensor Produc**T** **ATT**en**T**ion **T**ransformer (T6), a new model architecture for sequence modeling. Through extensive empirical evaluation of language modeling tasks, we demonstrate that T6 exceeds the performance of standard Transformer baselines including MHA, MQA, GQA, and MLA across various metrics, including perplexity and a range of renowned evaluation benchmarks. Notably, TPA's memory efficiency enables the processing of significantly longer sequences under fixed resource constraints, addressing a critical scalability challenge in modern language models. The code is available at https://anonymous.4open.science/r/T6-anonymous-2025.

## 1 Introduction

Large language models (LLMs) have revolutionized natural language processing, demonstrating exceptional performance across tasks (Brown et al., 2020; Chowdhery et al., 2023; Touvron et al., 2023; Bubeck et al., 2023). As these models evolve, their ability to process longer contexts becomes increasingly important for sophisticated applications such as document analysis, complex reasoning, and code

---
[1]Anonymous Institution, Anonymous City, Anonymous Region, Anonymous Country. Correspondence to: Anonymous Author <anon.email@domain.com>.

Preliminary work. Under review by the International Conference on Machine Learning (ICML). Do not distribute.

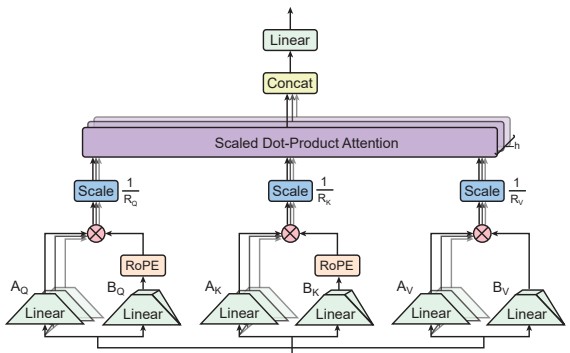

*Figure 1.* Tensor Product Attention (TPA) in the **T**ensor Produc**T** **ATT**en**T**ion **T**ransformer (T6). Different from multi-head attention, in each layer, firstly the hidden state goes through different linear layers to get the latent factor matrices $\mathbf{A}_{(\cdot)}$'s and $\mathbf{B}_{(\cdot)}$'s for query, key, and value. We additionally apply RoPE to $\mathbf{B}_Q$ and $\mathbf{B}_K$ for query and key. Then the multi-head query, key, and value vectors are attained by the tensor product of $\mathbf{A}_{(\cdot)}$ and $\mathbf{B}_{(\cdot)}$. Finally, the output of TPA is produced by scaled dot-product attention followed by linear projection of concatenated results of multiple heads.

completions. However, managing longer sequences during inference poses significant computational and memory challenges, particularly due to the storage of key-value (KV) caches (Zhang et al., 2023c; Liu et al., 2024c). Because memory consumption grows linearly with sequence length, the maximum context window is limited by practical hardware constraints.

A variety of solutions have been explored to address this memory bottleneck. Some approaches compress or selectively prune cached states through sparse attention patterns (Child et al., 2019) or token eviction strategies (Zhang et al., 2023c; Xiao et al., 2024; Ribar et al., 2024), though such methods risk discarding tokens that may later prove important. Other work proposes off-chip storage of key-value states (He & Zhai, 2024), at the expense of increased I/O latency. Attention variants like multi-query attention (MQA) (Shazeer, 2019) and grouped-query attention (GQA) (Ainslie et al., 2023) reduce per-token cache requirements by sharing keys and values across heads, but often compromise flexibility or require significant architectural modifications. Meanwhile, low-rank weight factorization methods such as LoRA (Hu et al., 2022) effectively reduce fine-tuning memory, yet do not address the KV cache overhead that dominates runtime. The recently introduced

Multi-head Latent Attention (MLA) in Deepseek-V2 (Liu et al., 2024a) caches compressed key-value representations but needs additional position-encoded parameters per head due to incompatibility with Rotary Position Embedding (RoPE) efficiently (Su et al., 2024).

In order to overcome the limitations of existing approaches, we introduce *Tensor Product Attention* (TPA), as illustrated in Figure 1, a novel architecture that uses higher-order tensors to factorize queries (Q), keys (K), and values (V) during attention computation. By dynamically factorizing *activations* rather than static weights (e.g., LoRA), TPA constructs low-rank, contextual representations that substantially reduce KV cache memory usage with improved representational capacity. In practice, TPA can reduce the memory overhead by an order of magnitude compared to standard multi-head attention (MHA) with lower pretraining validation loss (perplexity) and improved downstream performance.

A key advantage of TPA is its native compatibility with rotary positional embeddings (RoPE) (Su et al., 2024), enabling a straightforward drop-in replacement for multi-head attention (MHA) layers in modern LLM architectures such as LLaMA (Touvron et al., 2023) and Gemma (Team et al., 2024).

Our primary contributions are summarized as follows:

- We propose **Tensor Product Attention (TPA)**, A mechanism that factorizes **Q**, **K**, and **V** activations using *contextual* tensor-decompositions to achieve $10\times$ or more reduction in inference-time KV cache size relative to standard attention mechanism (Vaswani et al., 2017) with improved performance compared to previous methods such as MHA, MQA, GQA, and MLA. In addition, we **unify existing attention mechanisms** by revealing that MHA, MQA, and GQA *all* arise naturally as non-contextual variants of TPA.

- We introduce the **T**ensor Produc**T** A**TT**en**T**ion **T**ransformer (T6), a new TPA-based model architecture for sequence modeling. On language modeling experiments, T6 consistently improves validation perplexity and downstream evaluation performance with reduced KV cache size.

- We show **TPA** integrates seamlessly with RoPE (Su et al., 2024), facilitating easy adoption in popular foundation model architectures such as LLaMA and Gemma.

## 2 Background

In this section, we review two classical forms of attention: Scaled Dot-Product Attention, and Multi-Head Attention (MHA) (Vaswani et al., 2017). More types of attention are introduced in the Appendix E, including Multi-Query Attention (MQA) (Shazeer, 2019), and Grouped Query Attention (GQA) (Ainslie et al., 2023), as well as a recent method called Multi-head Latent Attention (MLA) used in DeepSeek-V2 (Liu et al., 2024a) and DeepSeek-V3 (Liu et al., 2024b). We also introduce Rotary Position Embedding (RoPE, Su et al. (2024)), which is commonly used in recent works of large language models.

**Notations.** We use bold uppercase letters (e.g., $\mathbf{X}$, $\mathbf{Q}$) for matrices, bold lowercase (e.g., $\mathbf{a}$, $\mathbf{b}$) for vectors, and italic uppercase (e.g., $\boldsymbol{W}_i^Q$) for learnable parameter matrices. We denote by $[n]$ the set $\{1, \ldots, n\}$ for some positive integer $n$. We use $\top$ to denote the transpose of a vector or a matrix. Let $d_{\text{model}}$ be the embedding dimension, $h$ the number of attention heads, $d_h$ the dimension per head, $\mathbf{x}_t \in \mathbb{R}^d$ the input for the $t$-th token at a given attention layer, $\mathbf{X} \in \mathbb{R}^{T \times d_{\text{model}}}$ denotes the input embeddings for $T$ tokens, and $\mathbf{Q}, \mathbf{K}, \mathbf{V} \in \mathbb{R}^{T \times h \times d_h}$ denote the queries, keys, and values of $h$ heads for $T$ tokens. With a little abuse of notation, $\mathbf{Q}_i$, $\mathbf{K}_i$, $\mathbf{V}_i \in \mathbb{R}^{T \times d_h}$ denote the $i$-th head of queries, keys, and values, and $\mathbf{Q}_t$, $\mathbf{K}_t$, $\mathbf{V}_t \in \mathbb{R}^{h \times d_h}$ denote the heads of the query, key, and value for $t$-th token.

Throughout the paper, $\boldsymbol{W}^Q, \boldsymbol{W}^K, \boldsymbol{W}^V$ denote projection matrices for queries, keys, and values, respectively. In multi-head attention, each head is associated with its own set of $\boldsymbol{W}_i^Q, \boldsymbol{W}_i^K, \boldsymbol{W}_i^V$, and each has dimension $\boldsymbol{W}_i^Q, \boldsymbol{W}_i^K, \boldsymbol{W}_i^V \in \mathbb{R}^{d_{\text{model}} \times d_k}$, where $d_k$ is typically set to $d_h$, the dimension of each head.[5] Similarly, we have an output projection matrix $\boldsymbol{W}^O \in \mathbb{R}^{(h \cdot d_h) \times d_{\text{model}}}$. For methods like MQA and GQA, some of these are shared or partially shared across heads, but their shapes remain consistent.

We define the tensor product of two vectors as follows: for vectors $\mathbf{a} \in \mathbb{R}^m, \mathbf{b} \in \mathbb{R}^n$, the tensor product of $\mathbf{a}$ and $\mathbf{b}$ is:

$$\mathbf{a} \otimes \mathbf{b} = \mathbf{C} \in \mathbb{R}^{m \times n}, \text{with } C_{ij} = a_i b_j,$$

where $a_i$ and $b_j$ are the $i$-th and $j$-th elements of $\mathbf{a}$ and $\mathbf{b}$ respectively, and $C_{ij}$ is the $(i, j)$-th entry of $\mathbf{C}$. We also define the vectorization of a matrix $\mathbf{C} \in \mathbb{R}^{m \times n}$ by:

$$\text{vec}(\mathbf{C}) = \mathbf{d} \in \mathbb{R}^{mn}, \text{with } d_{i \cdot n + j} = C_{ij},$$

where $d_{i \cdot n + j}$ is the $(i \cdot n + j)$-th element of $\mathbf{d}$.

### 2.1 Scaled Dot-Product Attention

Scaled dot-product attention (Vaswani et al., 2017) determines how to focus on different parts of an input sequence by comparing queries ($\mathbf{Q}$) and keys ($\mathbf{K}$). It produces a weighted combination of the values ($\mathbf{V}$). Formally, the attention output is:

$$\text{Attention}(\mathbf{Q}, \mathbf{K}, \mathbf{V}) = \text{Softmax}\left(\frac{\mathbf{Q}\mathbf{K}^\top}{\sqrt{d_k}}\right)\mathbf{V},$$

where each of $\mathbf{Q}, \mathbf{K}, \mathbf{V}$ is an $(n \times d_k)$ matrix for $n$ tokens and key dimension $d_k$. The division by $\sqrt{d_k}$ stabilizes training by controlling the scale of the inner products.

---

[5]Often, one sets $h \times d_h = d_{\text{model}}$, so each head has query/key/value dimension $d_h$.

## 2.2 Multi-Head Attention (MHA)

Multi-Head Attention (MHA) extends scaled dot-product attention by dividing the model's internal representation into several *heads*. Each head learns different projections for queries, keys, and values, allowing the model to attend to different types of information. For each token embedding $\mathbf{x}_t \in \mathbb{R}^{d_{\text{model}}}$, MHA computes each head $i$ as follows:

$$\mathbf{Q}_{t,i} = (\boldsymbol{W}_i^Q)^\top \mathbf{x}_t \in \mathbb{R}^{d_h},$$
$$\mathbf{K}_{t,i} = (\boldsymbol{W}_i^K)^\top \mathbf{x}_t \in \mathbb{R}^{d_h},$$
$$\mathbf{V}_{t,i} = (\boldsymbol{W}_i^V)^\top \mathbf{x}_t \in \mathbb{R}^{d_h},$$
$$\mathbf{head}_i = \text{Attention}\Big(\mathbf{Q}_i, \mathbf{K}_i, \mathbf{V}_i\Big),$$

where $\boldsymbol{W}_i^Q, \boldsymbol{W}_i^K, \boldsymbol{W}_i^V \in \mathbb{R}^{d_{\text{model}} \times d_h}$ are learnable projection matrices for the $i$-th head, $\mathbf{Q}_i, \mathbf{K}_i, \mathbf{V}_i \in \mathbb{R}^{T \times d_h}$. After computing each head's attention, the outputs are concatenated and mapped back to the original dimension via another matrix $\boldsymbol{W}^O \in \mathbb{R}^{h d_h \times d_{\text{model}}}$:

$$\text{MHA}(\mathbf{Q}, \mathbf{K}, \mathbf{V}) = \text{Concat}\big(\mathbf{head}_1, \ldots, \mathbf{head}_h\big) \boldsymbol{W}^O.$$

MHA can capture a rich set of dependencies while each head focuses on different subspaces.

## 2.3 Rotary Position Embedding (RoPE)

Many recent LLMs use rotary position embedding (RoPE; Su et al., 2024) to encode positional information in the query/key vectors. Specifically, let $\text{RoPE}_t$ denote the rotation operator $\mathbf{T}_t \in \mathbb{R}^{d_h \times d_h}$ corresponding to the $t$-th position. $\mathbf{T}_t$ is a block-diagonal matrix, which consists of block-diagonal matrix $\begin{pmatrix} \cos(t\theta_j) & -\sin(t\theta_j) \\ \sin(t\theta_j) & \cos(t\theta_j) \end{pmatrix}, j \in \{1, \cdots, d_h/2\}$, where $\{\theta_j\}$ are pre-defined frequency parameters, e.g., $\theta_j = 1/10000^{2j/d_h}$. Then we define

$$\text{RoPE}\left(\mathbf{Q}_t\right) \triangleq \mathbf{Q}_t \mathbf{T}_t, \quad \text{where } \mathbf{Q}_t \in \mathbb{R}^{h \times d_h}.$$

A fundamental property is that

$$\mathbf{T}_t \, \mathbf{T}_s^\top = \mathbf{T}_{t-s}, \tag{2.1}$$

which ensures that relative positions $(t - s)$ are preserved, thereby providing a form of translation invariance in the rotary position embedding.

## 3 Tensor Product Attention

In this section, we provide a detailed description of our proposed *Tensor Product Attention* (TPA), which allows *contextual* low-rank factorization for queries, keys, and values. First, we explain how TPA factorizes queries, keys, and values with explicit tensor shapes. Next, we describe how TPA can be integrated into the multi-head attention framework and how it reduces memory consumption in KV caching at inference time. Finally, we show how RoPE can seamlessly integrate with TPA (including a pre-rotated variant).

## 3.1 Tensor Factorization of Queries, Keys, and Values

Let $\mathbf{x}_t \in \mathbb{R}^{d_{\text{model}}}$ for $t = 1, \ldots, T$ be the hidden-state vector corresponding to the $t$-th token in a sequence of length $T$. A typical multi-head attention block has $h$ heads, each of dimension $d_h$, satisfying $d_{\text{model}} = h \times d_h$. Standard attention projects the entire sequence into three tensors, $\mathbf{Q}, \mathbf{K}, \mathbf{V} \in \mathbb{R}^{T \times h \times d_h}$, where $\mathbf{Q}_t, \mathbf{K}_t, \mathbf{V}_t \in \mathbb{R}^{h \times d_h}$ denote the slices for the $t$-th token.

**Contextual Factorization (CF).** Instead of forming each head's query, key, or value via a single linear map, TPA factorizes each $\mathbf{Q}_t, \mathbf{K}_t, \mathbf{V}_t$ into a sum of (contextual) tensor products whose ranks are $R_q$, $R_k$, and $R_v$, respectively and may differ. Specifically, for each token $t$, with a small abuse of notation, we define:

$$\mathbf{Q}_t = \frac{1}{R_Q} \sum_{r=1}^{R_Q} \mathbf{a}_r^Q(\mathbf{x}_t) \otimes \mathbf{b}_r^Q(\mathbf{x}_t), \tag{3.1}$$

$$\mathbf{K}_t = \frac{1}{R_K} \sum_{r=1}^{R_K} \mathbf{a}_r^K(\mathbf{x}_t) \otimes \mathbf{b}_r^K(\mathbf{x}_t), \tag{3.2}$$

$$\mathbf{V}_t = \frac{1}{R_V} \sum_{r=1}^{R_V} \mathbf{a}_r^V(\mathbf{x}_t) \otimes \mathbf{b}_r^V(\mathbf{x}_t), \tag{3.3}$$

where $\mathbf{a}_r^Q(\mathbf{x}_t), \mathbf{a}_r^K(\mathbf{x}_t), \mathbf{a}_r^V(\mathbf{x}_t) \in \mathbb{R}^h$, $\mathbf{b}_r^Q(\mathbf{x}_t), \mathbf{b}_r^K(\mathbf{x}_t), \mathbf{b}_r^V(\mathbf{x}_t) \in \mathbb{R}^{d_h}$. Hence, for queries, each tensor product $\mathbf{a}_r^Q(\mathbf{x}_t) \otimes \mathbf{b}_r^Q(\mathbf{x}_t) \colon \mathbb{R}^h \times \mathbb{R}^{d_h} \to \mathbb{R}^{h \times d_h}$ adds up to form the query slice $\mathbf{Q}_t \in \mathbb{R}^{h \times d_h}$. Similarly, analogous definitions apply to key slice $\mathbf{K}_t$ and value slice $\mathbf{V}_t$.

**Latent Factor Maps.** Each factor in the tensor product depends on the token's hidden state $\mathbf{x}_t$. For example, for queries, we can write:

$$\mathbf{a}_r^Q(\mathbf{x}_t) = \boldsymbol{W}_r^{a^Q} \mathbf{x}_t \in \mathbb{R}^h, \quad \mathbf{b}_r^Q(\mathbf{x}_t) = \boldsymbol{W}_r^{b^Q} \mathbf{x}_t \in \mathbb{R}^{d_h},$$

and similarly for keys and values.

One often merges the rank index into a single output dimension. For instance, for queries:

$$\mathbf{a}^Q(\mathbf{x}_t) = \boldsymbol{W}^{a^Q} \mathbf{x}_t \in \mathbb{R}^{R_q \cdot h}, \; \mathbf{b}^Q(\mathbf{x}_t) = \boldsymbol{W}^{b^Q} \mathbf{x}_t \in \mathbb{R}^{R_q \cdot d_h},$$

which are then reshaped into $\mathbf{A}_Q(\mathbf{x}_t) \in \mathbb{R}^{R_q \times h}$ and $\mathbf{B}_Q(\mathbf{x}_t) \in \mathbb{R}^{R_q \times d_h}$. Summing over $R_q$ and scaled by $\frac{1}{R_q}$ yields

$$\mathbf{Q}_t = \frac{1}{R_Q} \mathbf{A}_Q(\mathbf{x}_t)^\top \mathbf{B}_Q(\mathbf{x}_t) \in \mathbb{R}^{h \times d_h}.$$

Repeating for all tokens reconstitutes $\mathbf{Q} \in \mathbb{R}^{T \times h \times d_h}$. Similarly, procedures can be applied to obtain $\mathbf{K}$ and $\mathbf{V}$ with ranks $R_k$ and $R_v$, respectively.

**Scaled Dot-Product Attention.** Once $\mathbf{Q}, \mathbf{K}, \mathbf{V}$ are factorized, multi-head attention proceeds as in standard Transformers. For each head $i \in \{1, \ldots, h\}$:

$$\mathbf{head}_i = \mathrm{Softmax}\left(\frac{1}{\sqrt{d_h}} \mathbf{Q}_i (\mathbf{K}_i)^\top\right) \mathbf{V}_i, \qquad (3.4)$$

where $\mathbf{Q}_i, \mathbf{K}_i, \mathbf{V}_i \in \mathbb{R}^{T \times d_h}$ are the slices along the head dimension. Concatenating these $h$ heads along the last dimension yields an $\mathbb{R}^{T \times (h \cdot d_h)}$ tensor, which is projected back to $\mathbb{R}^{T \times d_{\mathrm{model}}}$ by an output weight matrix $\boldsymbol{W}^O \in \mathbb{R}^{(h \cdot d_h) \times d_{\mathrm{model}}}$:

$$\mathrm{TPA}(\mathbf{Q}, \mathbf{K}, \mathbf{V}) = \mathrm{Concat}\big(\mathbf{head}_1, \ldots, \mathbf{head}_h\big) \boldsymbol{W}^O. \qquad (3.5)$$

**Parameter Initialization.** We initialize the weight matrices $\boldsymbol{W}_r^{a_Q}, \boldsymbol{W}_r^{a_K}, \boldsymbol{W}_r^{a_V}, \boldsymbol{W}_r^{b_Q}, \boldsymbol{W}_r^{b_K}, \boldsymbol{W}_r^{b_V}$ using Xavier initialization (Glorot & Bengio, 2010). Specifically, each entry of the weight matrix is drawn from a uniform distribution with bounds $[-\sqrt{6/(n_{\mathrm{in}} + n_{\mathrm{out}})}, \sqrt{6/(n_{\mathrm{in}} + n_{\mathrm{out}})}]$, where $n_{\mathrm{in}}$ and $n_{\mathrm{out}}$ are the input and output dimensions of the respective weight matrices. This initialization strategy helps maintain the variance of activations and gradients across the network.

### 3.2 RoPE Compatibility and Acceleration

In a typical workflow of adding RoPE to standard multi-head attention, one first computes $\mathbf{Q}_t, \mathbf{K}_s \in \mathbb{R}^{h \times d_h}$ of the $t$-th token and $s$-th token and then applies:

$$\mathbf{Q}_t \mapsto \widetilde{\mathbf{Q}}_t = \mathrm{RoPE}_t(\mathbf{Q}_t), \qquad \mathbf{K}_s \mapsto \widetilde{\mathbf{K}}_s = \mathrm{RoPE}_s(\mathbf{K}_s).$$

**Direct Integration.** A useful optimization is to integrate RoPE directly into the TPA factorization. For example, one can *pre-rotate* the token-dimension factors:

$$\widetilde{\mathbf{B}}_K(\mathbf{x}_t) \longleftarrow \mathrm{RoPE}_t\big(\mathbf{B}_K(\mathbf{x}_t)\big), \qquad (3.6)$$

yielding a *pre-rotated* key representation:

$$\widetilde{\mathbf{K}}_t = \frac{1}{R_K} \sum_{r=1}^{R_K} \mathbf{a}_{(r)}^K(\mathbf{x}_t) \otimes \mathrm{RoPE}_t\big(\mathbf{b}_{(s)}^K(\mathbf{x}_t)\big)$$

$$= \frac{1}{R_K} \mathbf{A}_K(\mathbf{x}_t)^\top \mathrm{RoPE}_t\big(\mathbf{B}_K(\mathbf{x}_t)\big).$$

Thus, each $\mathbf{K}_t$ is already rotated before caching, removing the need for explicit rotation at the decoding time and accelerating autoregressive inference. Depending on hardware and performance requirements, one can also adopt different RoPE integration approaches for training and inference.

**Theorem 1** (RoPE's Compatibility with TPA)**.** Let $\mathbf{Q}_t$ be factorized by TPA as

$$\mathbf{Q}_t = \frac{1}{R_Q} \mathbf{A}_Q(\mathbf{x}_t)^\top \mathbf{B}_Q(\mathbf{x}_t) \in \mathbb{R}^{h \times d_h},$$

where $\mathbf{A}_Q(\mathbf{x}_t) \in \mathbb{R}^{R_Q \times h}$ and $\mathbf{B}_Q(\mathbf{x}_t) \in \mathbb{R}^{R_Q \times d_h}$. Then we have:

$$\mathrm{RoPE}(\mathbf{Q}_t) = \frac{1}{R_Q} \mathbf{A}_Q(\mathbf{x}_t)^\top \widetilde{\mathbf{B}}_Q(\mathbf{x}_t), \qquad (3.7)$$

where $\widetilde{\mathbf{B}}_Q(\mathbf{x}_t) = \mathrm{RoPE}_t\big(\mathbf{B}_Q(\mathbf{x}_t)\big)$. In addition, assume $\mathbf{Q}_t$ and $\mathbf{K}_s$ are factorized by TPA and then rotated by $\mathrm{RoPE}_t, \mathrm{RoPE}_s$. Let $\widetilde{\mathbf{Q}}_t = \mathrm{RoPE}_t(\mathbf{Q}_t)$ and $\widetilde{\mathbf{K}}_s = \mathrm{RoPE}_s(\mathbf{K}_s)$. Then we have

$$\mathrm{RoPE}_{t-s}(\mathbf{Q}_t)\mathbf{K}_s^\top = \widetilde{\mathbf{Q}}_t \widetilde{\mathbf{K}}_s^\top,$$

Focusing on individual heads $i$, the above matrix equality implies:

$$\mathrm{RoPE}_{t-s}\big(\mathbf{q}_{t,i}\big)^\top \mathbf{k}_{s,i} = \widetilde{\mathbf{q}}_{t,i}^\top \widetilde{\mathbf{k}}_{s,i}.$$

where $\mathbf{q}_{t,i} \in \mathbb{R}^{d_h}$ is the $i$-th query head of $t$-th token, and $\mathbf{k}_{s,i} \in \mathbb{R}^{d_h}$ is the $j$-th key head of $s$-th token, and

$$\widetilde{\mathbf{q}}_{t,i} = \mathrm{RoPE}(\mathbf{q}_{t,i}) = \mathbf{T}_t \mathbf{q}_{t,i} \in \mathbb{R}^{d_h}$$

$$\widetilde{\mathbf{k}}_{s,i} = \mathrm{RoPE}(\mathbf{k}_{s,i}) = \mathbf{T}_s \mathbf{k}_{s,i} \in \mathbb{R}^{d_h}.$$

Theorem 1 indicates that TPA does not break RoPE's relative translational property. We prove Theorem 1 in Appendix C.1. In short, $\mathrm{RoPE}_t$ acts as a block-diagonal orthogonal transform (i.e., a matrix $\mathbf{T}_t$) on $\mathbf{B}_Q(\mathbf{x}_t)$. Consequently, $\mathbf{A}_Q(\mathbf{x}_t)$ remains unchanged, while each column of $\mathbf{B}_Q(\mathbf{x}_t)$ is rotated appropriately, preserving the TPA structure.

### 3.3 KV Caching and Memory Reduction

In autoregressive decoding, standard attention caches $\mathbf{K}_t, \mathbf{V}_t \in \mathbb{R}^{h \times d_h}$ for each past token $t$. This accumulates to $\mathbb{R}^{T \times h \times d_h}$ for keys and $\mathbb{R}^{T \times h \times d_h}$ for values, i.e., $2 T h d_h$ total.

**TPA Factorized KV Caching.** Instead of storing the full $\mathbf{K}_t$ and $\mathbf{V}_t$, TPA stores only their factorized ranks. Specifically, we keep

$$\mathbf{A}_K(\mathbf{x}_t), \widetilde{\mathbf{B}}_K(\mathbf{x}_t) \quad \text{and} \quad \mathbf{A}_V(\mathbf{x}_t), \mathbf{B}_V(\mathbf{x}_t),$$

where $\mathbf{A}_K(\mathbf{x}_t) \in \mathbb{R}^{R_K \times h}$, $\widetilde{\mathbf{B}}_K(\mathbf{x}_t) \in \mathbb{R}^{R_K \times d_h}$, $\mathbf{A}_V(\mathbf{x}_t) \in \mathbb{R}^{R_V \times h}$, $\mathbf{B}_V(\mathbf{x}_t) \in \mathbb{R}^{R_V \times d_h}$. Hence, the memory cost per token is

$$\underbrace{R_K(h + d_h)}_{\text{for K}} + \underbrace{R_V(h + d_h)}_{\text{for V}} = (R_K + R_V)(h + d_h).$$

Compared to the standard caching cost of $2 h d_h$, the ratio is:

$$\frac{(R_K + R_V)(h + d_h)}{2 h d_h}.$$

For large $h$ and $d_h$ (typically $d_h = 64$ or $128$), setting $R_K, R_V \ll h$ (e.g., rank 1 or 2) often yields $10\times$ or more reduction.

*Table 1.* Comparison of different attention mechanisms. Here, $R_Q$, $R_K$, and $R_V$ denote the ranks for queries, keys, and values in TPA, respectively. Variants of TPA, such as TPA (KVonly), TPA (Non-contextual A), and TPA (Non-contextual B), are detailed in Section F. For MLA, $d_h^R$ and $d_h$ are the dimensions for RoPE and non-RoPE parts; $d_c'$ and $d_c$ are the dimensions of compressed vectors for query and key-value, respectively.

| METHOD | KV CACHE | # PARAMETERS | # QUERY HEADS | # KV HEADS |
|---|---|---|---|---|
| MHA | $2hd_h$ | $4d_{\text{model}}^2$ | $h$ | $h$ |
| MQA | $2d_h$ | $(2 + 2/h)d_{\text{model}}^2$ | $h$ | $1$ |
| GQA | $2gd_h$ | $(2 + 2g/h)d_{\text{model}}^2$ | $h$ | $g$ |
| MLA | $d_c + d_h^R$ | $d_c'(d_{\text{model}} + hd_h + hd_h^R)$ $+d_{\text{model}}d_h^R + d_c(d_{\text{model}} + 2hd_h)$ | $h$ | $h$ |
| TPA | $(R_K + R_V)(h + d_h)$ | $d_{\text{model}}(R_Q + R_K + R_V)(h + d_h) + d_{\text{model}}hd_h$ | $h$ | $h$ |
| TPA (KVonly) | $(R_K + R_V)(h + d_h)$ | $d_{\text{model}}(R_K + R_V)(h + d_h) + 2d_{\text{model}}hd_h$ | $h$ | $h$ |
| TPA (Non-contextual A) | $(R_K + R_V)d_h$ | $(R_Q + R_K + R_V)(d_{\text{model}}d_h + h) + d_{\text{model}}hd_h$ | $h$ | $h$ |
| TPA (Non-contextual B) | $(R_K + R_V)h$ | $(R_Q + R_K + R_V)(d_{\text{model}}h + d_h) + d_{\text{model}}hd_h$ | $h$ | $h$ |

### 3.4 Unifying MHA, MQA, and GQA as Non-contextual TPA

#### 3.4.1 MHA AS NON-CONTEXTUAL TPA

Standard multi-head attention (MHA) can be viewed as a specific instance of TPA in which: 1) the rank is set equal to the number of heads; 2) the head dimension factor is non-contextual (i.e., independent of the $t$-th token embedding $\mathbf{x}_t \in \mathbb{R}^{d_{\text{model}}}$); 3) the token dimension factor is a linear function of $\mathbf{x}_t$.

To match MHA with TPA, let $R_Q = R_K = R_V = h$. Focusing on $\mathbf{Q}_t$:

(a) **Non-contextual head factors.** Define

$$\mathbf{a}_i^Q = R_Q \mathbf{e}_i \in \mathbb{R}^h, \qquad (3.8)$$

where $\mathbf{e}_i \in \mathbb{R}^h$ is the $i$-th standard basis vector, so that $\mathbf{e}_i \otimes \cdot$ corresponds to the $i$-th head of $\mathbf{Q}_t$.

(b) **Contextual token factors.** Define

$$\mathbf{b}_i^Q(\mathbf{x}_t) = (\boldsymbol{W}_i^Q)^\top \mathbf{x}_t \in \mathbb{R}^{d_h}, \qquad (3.9)$$

where $\boldsymbol{W}_i^Q \in \mathbb{R}^{d_{\text{model}} \times d_h}$ is the per-head query projection defined before, hence $\mathbf{b}_i^Q(\mathbf{x}_t)$ dependent on $\mathbf{x}_t$.

Substituting (3.8)–(3.9) into (3.1) gives:

$$\mathbf{Q}_t = \sum_{i=1}^{h} \left[ \mathbf{e}_i \otimes \left((\boldsymbol{W}_i^Q)^\top \mathbf{x}_t\right) \right] \in \mathbb{R}^{h \times d_h}. \qquad (3.10)$$

Each term $\mathbf{e}_i \otimes \left((\boldsymbol{W}_i^Q)^\top \mathbf{x}_t\right)$ in (3.10) contributes only to the $i$-th row, reconstituting the usual MHA form of $\mathbf{Q}_t$. Analogous constructions hold for $\mathbf{K}_t$ and $\mathbf{V}_t$ using $\boldsymbol{W}_i^K, \boldsymbol{W}_i^V$. Thus, *MHA is a non-contextual, full-rank variant of TPA.*

**TPA with Non-contextual A.** More broadly, TPA can use non-contextual head-dimension factors $\mathbf{a}_r^Q, \mathbf{a}_r^K, \mathbf{a}_r^V \in \mathbb{R}^h$ (i.e., independent of $\mathbf{x}_t$), while allowing $\mathbf{b}_r^Q(\mathbf{x}_t), \mathbf{b}_r^K(\mathbf{x}_t), \mathbf{b}_r^V(\mathbf{x}_t)$ to remain context-dependent.

Then, for keys:

$$\mathbf{K}_t = \frac{1}{R_K} \sum_{r=1}^{R_K} \mathbf{a}_r^K \otimes \mathbf{b}_r^K(\mathbf{x}_t),$$

and similarly for queries/values. This reduces per-token computations and can be effective when head-dimension relationships are relatively stable across all tokens.

**MQA and GQA as Non-Contextual TPA.** Multi-Query Attention (MQA) (Shazeer, 2019) and Grouped Query Attention (GQA) (Ainslie et al., 2023)[6] also emerge naturally from TPA by restricting the head-dimension factors to be non-contextual *and* low-rank:

- **MQA as Rank-1 TPA.** In MQA, all heads share a *single* set of keys/values, corresponding to $R_K = R_V = 1$ along the head dimension. Concretely,

$$\mathbf{K}_t = (1, \ldots, 1)^\top \otimes \mathbf{b}^K(\mathbf{x}_t),$$
$$\mathbf{V}_t = (1, \ldots, 1)^\top \otimes \mathbf{b}^V(\mathbf{x}_t),$$

  forces every head to use the same $\mathbf{K}_t, \mathbf{V}_t$. Each head retains a distinct query projection, matching the MQA design.

- **GQA as Grouped Rank-1 TPA.** GQA partitions $h$ heads into $G$ groups, each sharing keys/values within that group. In TPA form, each group $g$ has a dedicated non-contextual factor pair $\mathbf{a}_g^K, \mathbf{a}_g^V \in \mathbb{R}^h$, which acts as a "mask" for the heads in that group. Varying $G$ from 1 to $h$ interpolates from MQA to standard MHA.

Hence, by constraining TPA's head-dimension factors to be constant masks (one for MQA; multiple for GQA), these popular variants are recovered as special cases.

### 3.5 Computational Cost.

For a detailed analysis of the computational cost of TPA, please refer to Appendix A, which shows that the training

---

[6]The original definitions of MQA and GQA are presented in Appendix E.1 and E.2, respectively.

and inference flops of TPA with optimized implementation (without materializing $\mathbf{Q}$, $\mathbf{K}$, and $\mathbf{V}$) are smaller than MHA, GQA, and MLA. Specifically, when we set $R_q = 6$, $R_k = R_v = 2$ (our default setting), TPA is $10\times$ or more faster on calculating $\mathbf{Q}\mathbf{K}^\top$ than MLA during inference (see Appendix A.8).

### 3.6 Model Architectures

We propose a new architecture called **T**ensor Produc**T** A**TT**en**T**ion **T**ransformer (T6), which uses our *Tensor Product Attention* (TPA) in place of standard MHA (multi-head attention) or GQA (grouped-query attention). Building upon the query, key, and value tensors $\mathbf{Q}, \mathbf{K}, \mathbf{V} \in \mathbb{R}^{T \times h \times d_h}$ defined in Section 3.1, T6 utilize the overall architecture of LLaMA (Touvron et al., 2023) while changing the self-attention block to our TPA-based version. The feed-forward network (FFN) adopts a SwiGLU layer, as in (Shazeer, 2020; Touvron et al., 2023).

**TPA QKV Factorization.** Let each token's hidden-state vector be $\mathbf{x}_t \in \mathbb{R}^{d_{\text{model}}}$, and we follow Section 3.1 to project the entire sequence into three tensors $\mathbf{Q}, \mathbf{K}, \mathbf{V} \in \mathbb{R}^{T \times h \times d_h}$, where $\mathbf{Q}_t$, $\mathbf{K}_t$, $\mathbf{V}_t \in \mathbb{R}^{h \times d_h}$ denote the slices for the $t$-th token. The factor components $\mathbf{a}_r^Q(\mathbf{x}_t), \mathbf{b}_r^Q(\mathbf{x}_t), \mathbf{a}_r^K(\mathbf{x}_t), \mathbf{b}_r^K(\mathbf{x}_t), \mathbf{a}_r^V(\mathbf{x}_t), \mathbf{b}_r^V(\mathbf{x}_t)$ are produced by linear transformations on $\mathbf{x}_t$. For instance, letting $\boldsymbol{W}_r^{a^Q} \in \mathbb{R}^{h \times d_{\text{model}}}$ and $\boldsymbol{W}_r^{b^Q} \in \mathbb{R}^{d_h \times d_{\text{model}}}$, we have:

$$\mathbf{a}_r^Q(\mathbf{x}_t) = \boldsymbol{W}_r^{a^Q}\,\mathbf{x}_t, \quad \mathbf{b}_r^Q(\mathbf{x}_t) = \boldsymbol{W}_r^{b^Q}\,\mathbf{x}_t.$$

In practice, we merge all ranks $r$ into a single dimension of the output, reshape, and sum over rank indices; see Section 3.1 for details. The factorization for K and V follows the same pattern.

**Rotary Positional Embedding (RoPE).** As discussed in Section 3.2, RoPE (Su et al., 2024) is applied to the $\mathbf{Q}$ and $\mathbf{K}$. Within TPA, we *pre-rotate* the factor $\mathbf{b}_t^Q(\mathbf{x}_t)$ and $\mathbf{b}_s^K(\mathbf{x}_s)$ directly, so that each $\mathbf{K}_s$ is already rotated prior to caching, see (3.6) and Theorem 1.

**Attention Step and Output Projection.** Once we have $\mathbf{Q}, \mathbf{K}, \mathbf{V}$ factorized per token with RoPE applied on $\mathbf{Q}$ and $\mathbf{K}$, the attention step proceeds for each head $i \in \{1, \ldots, h\}$ using (3.4). Finally, concatenating these $h$ heads and then projecting them back using an output weight matrix gives the final attention result, as shown in (3.5).

**SwiGLU Feed-Forward Network.** Following Shazeer (2020); Touvron et al. (2023), our T6 uses a SwiGLU-based Feed-Forward Network (FFN): $\text{FFN}(\mathbf{x}) = \left[\sigma(\mathbf{x}\,\boldsymbol{W}_1) \odot (\mathbf{x}\,\boldsymbol{W}_2)\right]\boldsymbol{W}_3$, where $\sigma$ is the SiLU (a.k.a., swish) nonlinearity, $\odot$ is element-wise product, and $\boldsymbol{W}_1, \boldsymbol{W}_2, \boldsymbol{W}_3$ are learnable parameters. Note that other activation functions can also be used.

**Overall T6 Block Structure.** Putting everything together, one T6 block consists of:

$$\mathbf{x} \leftarrow \mathbf{x} + \text{TPA}\big(\text{RMSNorm}(\mathbf{x})\big),$$
$$\mathbf{x} \leftarrow \mathbf{x} + \text{SwiGLU-FFN}\big(\text{RMSNorm}(\mathbf{x})\big).$$

We place norm layers (e.g., RMSNorm) before each sublayer. Stacking $L$ such blocks yields a T6 model architecture with $L$ layers.

## 4 Experiments

### 4.1 Language Modeling Tasks

All experiments reported in this paper are implemented on the `nanoGPT` code base (Karpathy, 2022), using the FineWeb-Edu 100B dataset (Lozhkov et al., 2024). The dataset contains 100 billion tokens for training and 0.1 billion tokens for validation. We compare T6 against the baseline Llama architecture (Touvron et al., 2023) with SwiGLU activation (Shazeer, 2020) and RoPE embeddings (Su et al., 2024), as well as Llama variants that replace Multi-Head Attention (MHA; Vaswani et al., 2017) with Multi-Query Attention (MQA; Shazeer, 2019), Grouped Query Attention (GQA; Ainslie et al., 2023), or Multi-head Latent Attention (MLA; Liu et al., 2024a). In our experiments, the number of heads $h$ is adjusted for each attention mechanism to ensure that all attention mechanisms have the same number of parameters as the standard Multi-Head Attention (MHA), which has $4d_{\text{model}}^2$ parameters per attention layer. We train models at four scales: *small* (124M parameters), *medium* (353M), *large* (773M), and *XL* (1.5B). Details on architecture hyperparameters and training hardware are shown in Appendix G.1.

**Training Setup.** We follow the `nanoGPT` training configuration. In particular, we use the AdamW (Loshchilov, 2017) optimizer with $(\beta_1, \beta_2) = (0.9, 0.95)$, a weight decay of 0.1, and gradient clipping at 1.0. We follow the same setting as `nanoGPT` that the learning rate is managed by a cosine annealing scheduler (Loshchilov & Hutter, 2016) with 2,000 warmup steps and a (total) global batch size of 480. For the *small*, *medium*, *large* and *XL* models, we set maximum learning rates of $6 \times 10^{-4}$, $3 \times 10^{-4}$, $2 \times 10^{-4}$, and $1 \times 10^{-4}$ (respectively), and minimum learning rates of $3 \times 10^{-5}$, $6 \times 10^{-5}$, $1 \times 10^{-5}$, and $1 \times 10^{-5}$ (respectively).

**Training & Validation Curves.** Figures 2 and 3 compare training and validation loss curves for the *medium* (353M), *large* (773M), and *XL* (1.5B) models on FineWeb-Edu-100B. Overall, **TPA** (red curves) and its simpler variant **TPA-KVonly** (pink curves) (see F) converge as fast as or faster than the baselines (MHA, MQA, GQA, MLA) while also achieving visibly lower final losses. For instance, in Figure 3(b), TPA and TPA-KVonly remain below the MHA baseline in terms of validation loss at nearly all training stages. Meanwhile, Multi-Head Latent Attention (MLA) (Liu et al., 2024a) (blue curves) generally trains

more slowly and yields higher losses.

**Validation Perplexity.** Figure 4 (in the Appendix) shows the validation perplexities of the *medium-* and *large*-scale models. Mirroring the loss curves, **TPA** and **TPA-KVonly** steadily outperform MHA, MQA, GQA, and MLA over the course of training. By the end of pretraining (around 49B tokens), TPA-based approaches achieve the lowest perplexities in most configurations.

**Downstream Evaluation.** We evaluate zero-shot and two-shot performance on standard benchmarks, including ARC (Yadav et al., 2019), BoolQ (Clark et al., 2019), HellaSwag (Zellers et al., 2019), OBQA (Mihaylov et al., 2018), PIQA (Bisk et al., 2020), WinoGrande (Sakaguchi et al., 2020) and MMLU (Hendrycks et al., 2021), using the `lm-evaluation-harness` codebase (Gao et al., 2024). For ARC-E, ARC-C, HellaSwag, OBQA, PIQA, and SciQ, we report accuracy norm; for other tasks, we report standard accuracy. Due to the page limitation, we only display the zero-shot evaluation results of *medium* and *large* models here in Tables 2 and 3. Zero-shot evaluation of *small* and *XL* models are displayed in Tables 6 and 7 in the appendix. Moreover, we also present 2-shot evaluation results in Tables 8, 9, 10 and 11 in the appendix.

For the *medium*-size (353M) models (Tables 2 and 9), TPA generally ties or outperforms all competing methods, achieving, for example, an average of 51.41% in zero-shot mode versus MHA's 50.11%, MQA's 50.44%, and MLA's 50.13%. When given two-shot prompts, TPA again leads with 53.12% average accuracy. A similar trend appears for the *large*-size (773M) models (Tables 3), where TPA-KVonly attains the highest average (53.52% zero-shot). And for the *XL* size (1.5B) models (Table 7), TPA-KVonly attains the highest average (55.03% zero-shot).

Our experiments confirm that TPA consistently matches or exceeds the performance of established attention mechanisms (MHA, MQA, GQA, MLA) across *medium* and *large* model scales. The fully factorized TPA excels on mid-scale models, while TPA-KVonly can rival or surpass it at larger scales. In both cases, factorizing the attention activations shrinks autoregressive KV cache requirements by up to $5\times$–$10\times$, thus enabling much longer context windows under fixed memory budgets. In summary, tensor product attention provides a flexible, memory-efficient alternative to standard multi-head attention, advancing the scalability of modern language models.

## 5 Related Work

**Transformers and Attention.** As a sequence-to-sequence architecture Transformer (Vaswani et al., 2017) introduced Multi-Head Attention (MHA), enabling more effective capture of long-range dependencies. Subsequent work has explored a variety of attention mechanisms aimed at improving scalability and efficiency, including sparse patterns (Child et al., 2019; Shi et al., 2023; Han et al., 2024; Liang et al., 2024a; Li et al., 2024; Liang et al., 2024b), kernel-based projections (Choromanski et al., 2021), and linearized transformers (Tsai et al., 2019; Katharopoulos et al., 2020; Schlag et al., 2021; Zhang et al., 2023b; Sun et al., 2023; Zhang et al., 2024). To decrease memory usage and circumvent the limitation of memory bandwidth in training, Shazeer (2019) proposed Multi-Query Attention (MQA) where multiple query heads share the same key head and value head. To tackle with the issue of quality degradation and instability in training, Grouped-Query Attention (GQA) (Ainslie et al., 2023) divides queries into several groups, and each group of queries shares a single key head and value head. Recently, DeepSeek-V2 (Liu et al., 2024a) applied multihead latent attention (MLA) to achieve better performance than MHA while reducing KV cache in inference time by sharing the same low-rank representation of key and value. Concurrently, Hu et al. (2024) proposed Multi-matrix Factorization Attention (MFA), which can be simply seen as MQA with low-rank factorized Q. Compared to the approaches above, TPA applied contextual tensor decompositions to represent queries, keys, and values activations compactly, achieving better reduction on the size of KV cache with improved performance.

**KV Cache Optimization.** During the inference time of Transformers, key and value tensors of the previous tokens are repeatedly computed due to their auto-regressive nature. To enhance efficiency, firstly proposed by Ott et al. (2019), these tensors can be cached in memory for future decoding, referred to as the KV cache. However, the KV cache requires additional memory usage and may add to more latencies due to the bandwidth limitation (Adnan et al., 2024). Therefore, previous studies have explored diverse approaches to mitigate these issues, including KV cache eviction to discard less significant tokens (Zhang et al., 2023c; Xiao et al., 2024; Cai et al., 2024; Adnan et al., 2024), dynamic sparse attention among selected keys and values (Ribar et al., 2024; Tang et al., 2024; Singhania et al., 2024), KV cache offloading to CPU (He & Zhai, 2024; Lee et al., 2024; Sun et al., 2024), as well as quantization of KV cache (Xiao et al., 2023; Liu et al., 2024c; Hooper et al., 2024). Different from the methods above, TPA reduces the size of the KV cache by using tensor-decomposed KV.

## 6 Conclusion

We introduced *Tensor Product Attention* (TPA), which factorizes query, key, and value matrices into rank-$R$ tensor products dependent on the token's hidden state. Storing only the factorized key/value components during autoregressive decoding substantially decreases the kv memory size with improved performance compared with MHA, MQA, GQA, and MLA. The approach is fully compatible with RoPE (and can store pre-rotated keys). Variants of TPA in-

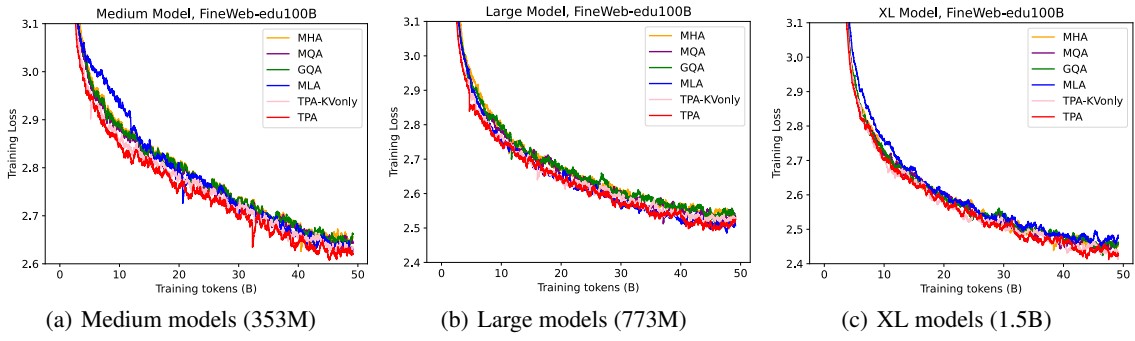

(a) Medium models (353M)   (b) Large models (773M)   (c) XL models (1.5B)

*Figure 2.* The training loss of medium-size (353M), large-size (773M) as well as XL-size (1.5B) models, with different attention mechanisms on the FineWeb-Edu 100B dataset.

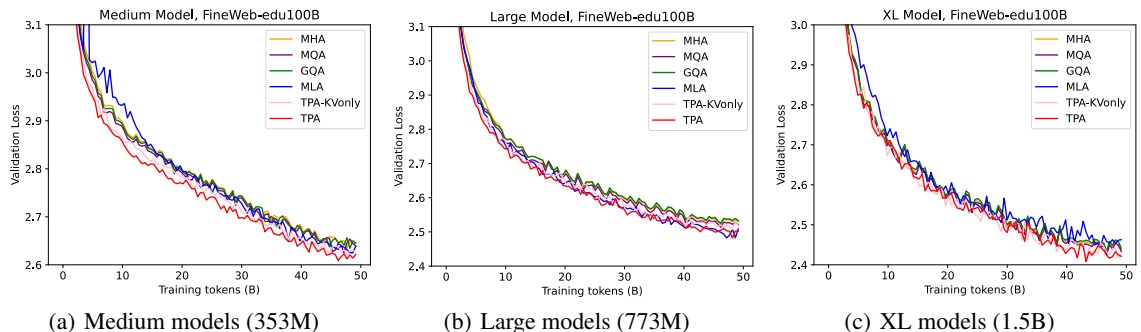

(a) Medium models (353M)   (b) Large models (773M)   (c) XL models (1.5B)

*Figure 3.* The validation loss of medium-size (353M), large-size (773M) as well as XL-size (1.5B) models, with different attention mechanisms on the FineWeb-Edu 100B dataset.

*Table 2.* The evaluation results of medium models with different attention mechanisms pre-trained using FineWeb-Edu 100B dataset (0-shot with lm-evaluation-harness). The best scores in each column are **bolded**. Abbreviations: HellaSw. = HellaSwag, W.G. = WinoGrande.

| Method | ARC-E | ARC-C | BoolQ | HellaSw. | OBQA | PIQA | W.G. | MMLU | SciQ | Avg. |
|---|---|---|---|---|---|---|---|---|---|---|
| MHA | **59.51** | 29.52 | 59.60 | 45.68 | 34.20 | 68.82 | 53.43 | 23.33 | 76.90 | 50.11 |
| MQA | 57.62 | **31.91** | 59.45 | 45.69 | 35.40 | 69.31 | 53.51 | **26.47** | 74.60 | 50.44 |
| GQA | 58.67 | 31.48 | 58.29 | 45.45 | 35.20 | 68.50 | **54.46** | 24.58 | 76.50 | 50.35 |
| MLA | 56.65 | 29.52 | 57.83 | 46.05 | 34.60 | 69.42 | 52.80 | 24.62 | 79.70 | 50.13 |
| **TPA-KVonly** | 58.01 | 30.12 | 58.01 | 45.95 | 35.60 | 69.10 | 53.12 | 25.39 | 75.10 | 50.04 |
| **TPA (non-ctx-A)** | 58.96 | 31.48 | **59.76** | 45.07 | 34.80 | 69.21 | 53.59 | 25.42 | 76.40 | 50.52 |
| **TPA** | 58.38 | 31.57 | 59.39 | **46.83** | **37.00** | **70.02** | 54.06 | 25.52 | **79.90** | **51.41** |

*Table 3.* The evaluation results of large models with different attention mechanisms pre-trained using the FineWeb-Edu 100B dataset (0-shot with lm-evaluation-harness). The best scores in each column are **bolded**. Abbreviations: HellaSw. = HellaSwag, W.G. = WinoGrande.

| Method | ARC-E | ARC-C | BoolQ | HellaSw. | OBQA | PIQA | W.G. | MMLU | SciQ | Avg. |
|---|---|---|---|---|---|---|---|---|---|---|
| MHA | 59.93 | 33.62 | 61.93 | 50.63 | 36.00 | 71.06 | 55.41 | 22.87 | 81.20 | 52.52 |
| MQA | 60.73 | 33.62 | 57.34 | 50.09 | 37.00 | 69.97 | 55.49 | 25.30 | 79.60 | 52.13 |
| GQA | 61.66 | 34.30 | 58.72 | 49.85 | 38.40 | 71.16 | 53.75 | 25.23 | 77.60 | 52.30 |
| MLA | **63.55** | 32.85 | 60.95 | **51.72** | **38.80** | 70.51 | 55.01 | 24.55 | **81.90** | 53.32 |
| **TPA-KVonly** | 63.26 | 34.13 | **61.96** | 50.66 | 37.20 | **72.09** | 55.25 | **26.06** | 81.10 | **53.52** |
| **TPA** | 63.22 | **35.58** | 60.03 | 51.26 | 36.80 | 71.44 | **55.56** | 24.77 | 79.60 | 53.10 |

clude factorizing only the key/value or sharing basis vectors across tokens. Overall, TPA offers a powerful mechanism for compressing KV storage while improving the model per-formance, thereby enabling longer sequence contexts under constrained memory.

## Impact Statement

This paper presents work whose goal is to advance the field of foundation models especially Large Language Models (LLMs). We believe that our work contributes meaningfully to the field, specifically on advancing the efficiency in the inference stage of LLMs by reducing KV cache size. By reducing memory requirements, our method could enable the deployment of capable language models on more resource-constrained devices and in broader settings, opening new avenues for their application in various downstream tasks. Lower memory usage typically correlates with reduced energy consumption, potentially decreasing the environmental footprint of LLM inference. This advancement underscores the potential of LLMs architecture design in both technological and societal contexts.

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

# Appendix

# A   Toward Faster Computation Without Materializing $\mathbf{Q}$, $\mathbf{K}$ and $\mathbf{V}$

We now explore whether it is possible to compute attention scores $\mathbf{Q}\,\mathbf{K}^\top$ of Tensor Product Attention (TPA) *directly* from their factorized forms, thereby reducing floating-point operations.

## A.1   Single-Head Factorization Setup Without Materializing $\mathbf{Q}$ and $\mathbf{K}$

Consider a single head $i$. Each query vector $\mathbf{Q}_t^{(i)} \in \mathbb{R}^{d_h}$ is factorized (with rank $R_q$):

$$\mathbf{Q}_t^{(i)} = \sum_{r=1}^{R_q} a_{q,i}^{(r)}(\mathbf{x}_t)\,\mathbf{b}_q^{(r)}(\mathbf{x}_t),$$

and each key vector $\mathbf{K}_\tau^{(i)} \in \mathbb{R}^{d_h}$ is factorized (with rank $R_k$):

$$\mathbf{K}_\tau^{(i)} = \sum_{s=1}^{R_k} a_{k,i}^{(s)}(\mathbf{x}_\tau)\,\mathbf{b}_k^{(s)}(\mathbf{x}_\tau).$$

Their dot-product for tokens $t, \tau$ is

$$\left[\mathbf{Q}^{(i)}\,(\mathbf{K}^{(i)})^\top\right]_{t,\tau} = \sum_{r=1}^{R_q}\sum_{s=1}^{R_k} a_{q,i}^{(r)}(\mathbf{x}_t)\,a_{k,i}^{(s)}(\mathbf{x}_\tau)\,\left\langle\mathbf{b}_q^{(r)}(\mathbf{x}_t), \mathbf{b}_k^{(s)}(\mathbf{x}_\tau)\right\rangle. \tag{A.1}$$

## A.2   Multi-Head Case

For multi-head attention with $h$ heads, one repeats the factorization across all heads. The $\mathbf{b}_q^{(r)}, \mathbf{b}_k^{(s)}$ vectors are shared across heads.

## A.3   Complexity Analysis

We compare the cost of standard multi-head attention versus TPA under two scenarios:

1. **Naïve:** Materialize $\mathbf{Q}$ and $\mathbf{K}$ from factors, then perform the usual batched GEMM.

2. **Specialized:** Attempt to compute $\mathbf{Q}\,\mathbf{K}^\top$ directly from the rank-$(R_q, R_k)$ factors without explicitly forming $\mathbf{Q}, \mathbf{K}$.

**Standard Multi-Head Attention.**   For batch size $B$ and sequence length $T$:

- *Projection cost:* $\mathcal{O}\!\left(B\,T\,d_{\text{model}}^2\right)$ or $\mathcal{O}\!\left(B\,T\,d_{\text{model}}\,d_h\right)$.

- *Dot-product:* $\mathbf{Q}\,(\mathbf{K})^\top \in \mathbb{R}^{(B\,h)\times T\times T}$ costs $\mathcal{O}\!\left(B\,T^2\,d_{\text{model}}\right)$.

For large $T$, the $\mathcal{O}(B\,T^2\,d_{\text{model}})$ term dominates.

**TPA: Naïve Implementation.**

- *Constructing factors:* $\mathcal{O}\!\left(B\,T\,d_{\text{model}} \times R_q(h + d_h) + R_k(h + d_h) + R_v(h + d_h)\right)$.

- *Materializing* $\mathbf{Q}, \mathbf{K}$: $\mathcal{O}\!\left(B\,T\,(R_q\,h\,d_h + R_k\,h\,d_h)\right)$.

- *Dot-product* $\mathbf{Q}\,(\mathbf{K})^\top$: $\mathcal{O}\!\left(B\,T^2\,d_{\text{model}}\right)$.

Typically $R_q, R_k, R_v \ll h$, so the overhead of constructing factors is small relative to $\mathcal{O}(T^2\,d_{\text{model}})$. Meanwhile, we still gain KV caching benefits.

**TPA: Specialized Implementation.**   If we bypass explicitly forming $\mathbf{Q}, \mathbf{K}$, each dot product $\mathbf{Q}_t \cdot \mathbf{K}_\tau$ is a double sum over rank indices. Below we detail its complexity.

### A.4 Complexity Analysis for the Specialized Implementation

**Single-Head Complexity.** A single attention head of dimension $d_h$. For each query:

$$\mathbf{Q}_t^{(i)} = \sum_{r=1}^{R_q} a_{q,i}^{(r)}(\mathbf{x}_t)\, \mathbf{b}_q^{(r)}(\mathbf{x}_t),$$

and for each key:

$$\mathbf{K}_\tau^{(i)} = \sum_{s=1}^{R_k} a_{k,i}^{(s)}(\mathbf{x}_\tau)\, \mathbf{b}_k^{(s)}(\mathbf{x}_\tau).$$

Their dot product:

$$\mathbf{Q}_t^{(i)} \cdot \mathbf{K}_\tau^{(i)} = \sum_{r=1}^{R_q}\sum_{s=1}^{R_k} \Big[ a_{q,i}^{(r)}(\mathbf{x}_t)\, a_{k,i}^{(s)}(\mathbf{x}_\tau) \Big] \, \big\langle \mathbf{b}_q^{(r)}(\mathbf{x}_t), \mathbf{b}_k^{(s)}(\mathbf{x}_\tau) \big\rangle.$$

For each pair $(r, s)$, we pay:

1. $\mathcal{O}(1)$ for multiplying two scalars,

2. $\mathcal{O}(d_h)$ for the dot product $\mathbf{b}_q^{(r)}(\mathbf{x}_t) \cdot \mathbf{b}_k^{(s)}(\mathbf{x}_\tau)$.

Since $(r, s)$ runs over $R_q \times R_k$, each token-pair $(t, \tau)$ costs roughly

$$\mathcal{O}\Big( R_q\, R_k\, \big(1 + d_h\big) \Big) \approx \mathcal{O}(R_q\, R_k\, d_h).$$

For $T$ queries and $T$ keys, that is $\mathcal{O}(T^2\, R_q\, R_k\, d_h)$ for a single head.

**Multi-Head and Batches (Reusing b-Dot Products).** When extending to $h$ heads, each head $i$ has its own scalar factors $a_{q,i}^{(r)}(\mathbf{x}_t)$ and $a_{k,i}^{(s)}(\mathbf{x}_\tau)$, but the b-vectors $\mathbf{b}_q^{(r)}(\mathbf{x}_t)$ and $\mathbf{b}_k^{(s)}(\mathbf{x}_\tau)$ can still be *shared* across all heads (assuming the same rank-$R$ factors for every head). Hence, one can split the total cost into two stages:

1. **b-Dot-Product Stage:**
   For each token pair $(t, \tau)$ and each rank pair $(r, s)$, compute the dot product

   $$\big\langle \mathbf{b}_q^{(r)}(\mathbf{x}_t),\ \mathbf{b}_k^{(s)}(\mathbf{x}_\tau) \big\rangle \in \mathbb{R}.$$

   Since each dot product is $\mathcal{O}(d_h)$ and there are $R_q R_k$ rank pairs as well as $T^2$ token pairs, this stage costs:

   $$\mathcal{O}\big(T^2\, R_q R_k\, d_h\big).$$

   Crucially, these b-dot products need only be computed *once* and can be cached for reuse by all heads.

2. **Per-Head Scalar Multiplications:**
   After the b-dot products are precomputed (and cached), each head $i$ only needs to multiply each stored dot product by the corresponding scalars $a_{q,i}^{(r)}(\mathbf{x}_t)\, a_{k,i}^{(s)}(\mathbf{x}_\tau)$. Since this scalar multiplication is $\mathcal{O}(1)$ per pair, and there are $T^2$ token pairs and $R_q R_k$ rank pairs for each of the $h$ heads, this step costs:

   $$\mathcal{O}\big(h\, T^2\, R_q R_k\big).$$

Putting these together, for batch size $B$, the total cost is

$$\mathcal{O}\big(B\, T^2\, R_q R_k\, d_h\big) + \mathcal{O}\big(B\, T^2\, h\, R_q R_k\big) = \mathcal{O}\Big(B\, T^2\, R_q R_k\big(d_h + h\big)\Big).$$

By contrast, the standard multi-head attention dot-product step is $\mathcal{O}\big(B\, T^2\, h\, d_h\big)$. Hence, for the specialized TPA approach to *reduce* flops,

$$R_q R_k\, (d_h + h) \leq h\, d_h.$$

Thus a practical guideline is to ensure $R_q R_k < h\, \frac{d_h}{d_h + h}$. When that holds, bypassing explicit materialization of $\mathbf{Q}$ and $\mathbf{K}$ can be beneficial.

### A.5 Toward Faster Computation Without Materializing Q, K, V

We have explored a two-step procedure for computing $\mathbf{Q}\,\mathbf{K}^\top$ directly from factorized queries and keys *without* materializing $\mathbf{Q}$ or $\mathbf{K}$. Here, we extend this idea to also avoid explicitly forming $\mathbf{V}$. That is, all three activations $\mathbf{Q}, \mathbf{K}, \mathbf{V}$ remain factorized throughout the attention pipeline. We present a single-head formulation below, and then discuss multi-head and batch extensions.

**Extending the Two-Step Approach to Avoid V Materialization.** After we obtain $\mathbf{Q}\mathbf{K}^\top$, we apply $\alpha_{t,\tau} = \text{softmax}\big(\frac{1}{\sqrt{d_h}}\,(QK^\top)_{t,\tau}\big)$. The final attention output at token $t$ (single head) is

$$\text{head}(t) \;=\; \sum_{\tau=1}^{T} \alpha_{t,\tau}\,\mathbf{V}_\tau.$$

Using the factorization $\mathbf{V}_\tau = \sum_{u=1}^{R_v} a_v^{(u)}(\mathbf{x}_\tau)\,\mathbf{b}_v^{(u)}(\mathbf{x}_\tau)$, we write:

$$\text{head}(t) \;=\; \sum_{\tau=1}^{T} \alpha_{t,\tau} \sum_{u=1}^{R_v} a_v^{(u)}(\mathbf{x}_\tau)\,\mathbf{b}_v^{(u)}(\mathbf{x}_\tau).$$

Rearrange sums:

$$\text{head}(t) \;=\; \sum_{u=1}^{R_v}\Big[\sum_{\tau=1}^{T}\big(\alpha_{t,\tau}\,a_v^{(u)}(\mathbf{x}_\tau)\big)\,\mathbf{b}_v^{(u)}(\mathbf{x}_\tau)\Big].$$

We *still* do not explicitly form $\mathbf{V}_\tau$. Instead:

**Stage 1: Calculating $\mathbf{b}_v^{(u)}(\mathbf{x}_\tau)$ for all tokens.** We simply observe that each output $\text{head}(t)$ can be computed by summing vectors $\mathbf{b}_v^{(u)}(\mathbf{x}_\tau) \in \mathbb{R}^{d_h}$ weighted by $\alpha_{t,\tau}\,a_v^{(u)}(\mathbf{x}_\tau)$. The complexity for constructing $\mathbf{b}_v^{(u)}(\mathbf{x}_\tau)\,\forall u,\tau$ is $\mathcal{O}(T\,R_v\,d_h)$.

**Stage 2: Weighted Summation by $\alpha_{t,\tau}\,a_v^{(u)}(\mathbf{x}_\tau)$.** For each token $t$, the final attention head output is

$$\sum_{\tau=1}^{T}\alpha_{t,\tau}\sum_{u=1}^{R_v}a_v^{(u)}(\mathbf{x}_\tau)\,\mathbf{b}_v^{(u)}(\mathbf{x}_\tau) \;=\; \sum_{u=1}^{R_v}\Big[\sum_{\tau=1}^{T}\big(\alpha_{t,\tau}\,a_v^{(u)}(\mathbf{x}_\tau)\big)\,\mathbf{b}_v^{(u)}(\mathbf{x}_\tau)\Big].$$

We still never explicitly materialize $\mathbf{V}$. Instead, for each pair $(t,u)$, we must accumulate the sum of $T$ vectors $\mathbf{b}_v^{(u)}(\mathbf{x}_\tau) \in \mathbb{R}^{d_h}$, each scaled by the scalar $\alpha_{t,\tau}\,a_v^{(u)}(\mathbf{x}_\tau)$. Because each vector is $d_h$-dimensional, each $(t,u)$ summation costs $\mathcal{O}(T\,d_h)$. Summed over $t = 1\ldots T$ and $u = 1\ldots R_v$, the total work is $\mathcal{O}(T^2\,R_v\,d_h)$ for the entire sequence.

In practice, one precomputes all $\mathbf{b}_v^{(u)}(\mathbf{x}_\tau)$ for $\tau = 1\ldots N$, so each accumulation can be implemented as a simple "scalar-times-vector add" in a tight loop. This cost is usually smaller than the $\mathbf{Q}\mathbf{K}^\top$ factorized cost if $R_v \ll d_h$.

### A.6 Overall Complexity for Single-Head

Combining the four bullet-point stages from above (ignoring smaller overheads like the softmax) yields:

(i) **QK b-Dot Product Stage:** $\mathcal{O}(T^2\,R_q\,R_k\,d_h)$.

(ii) **QK Scalar-Multiply Stage:** $\mathcal{O}(T^2\,R_q\,R_k)$.

(iii) **Computing $\mathbf{b}_v^{(u)}(\mathbf{x}_\tau)$ for all tokens:** $\mathcal{O}(T\,R_v\,d_h)$.

(iv) **Weighted Summation by $\alpha_{t,\tau}\,a_v^{(u)}(\mathbf{x}_\tau)$:** $\mathcal{O}(T^2\,R_v\,d_h)$.

Hence, for a single head, the total cost is:

$$\mathcal{O}\Big(T^2\,R_q\,R_k\,d_h \;+\; T^2\,R_q\,R_k \;+\; T\,R_v\,d_h \;+\; T^2\,R_v\,d_h\Big).$$

In many cases (especially for large $T$), the $\mathcal{O}(T^2)$ terms dominate, so one often focuses on

$$\mathcal{O}\Big(T^2\,R_q\,R_k\,d_h \;+\; T^2\,R_q\,R_k \;+\; T^2\,R_v\,d_h\Big).$$

## A.7 Multi-Head and Batch Extensions (Reuse of b-Dot Products)

When extending to $h$ heads and batch size $B$, all sequence-length-dependent terms are multiplied by $\sim B\,h$. However, crucial b-dot products can be shared across heads:

**QK b-Dot Products.** Since each head has distinct scalar factors $a_{q,i}$, $a_{k,i}$ but the same $\mathbf{b}_q^{(r)}$, $\mathbf{b}_k^{(s)}$ across heads, each pairwise dot product

$$\langle \mathbf{b}_q^{(r)}(\mathbf{x}_t),\, \mathbf{b}_k^{(s)}(\mathbf{x}_\tau)\rangle$$

is computed just once per batch. That cost remains

$$\mathcal{O}\big(B\,T^2\,R_q\,R_k\,d_h\big),$$

not multiplied by $h$. After caching these dot products, each of the $h$ heads pays $\mathcal{O}\big(B\,T^2\,h\,R_q\,R_k\big)$ total for the head-specific scalar multiplications (the "$\alpha_{t,\tau}$"–like factors).

**V b-Evaluations.** Likewise, the $\mathbf{b}_v^{(u)}$ factors are shared across heads (i.e. one set of $\mathbf{b}_v$-vectors for all heads). Hence, computing all $\mathbf{b}_v^{(u)}(\mathbf{x}_\tau)$ for $\tau = 1 \ldots T$ (across the batch) is a one-time cost:

$$\mathcal{O}\big(B\,T\,R_v\,d_h\big).$$

Then each head $i$ has its own scalar factors $a_{v,i}^{(u)}(\mathbf{x}_\tau)$, so the final accumulation $\sum_{\tau=1}^{T} \alpha_{t,\tau}\, a_{v,i}^{(u)}(\mathbf{x}_\tau)\,\mathbf{b}_v^{(u)}(\mathbf{x}_\tau)$ costs $\mathcal{O}\big(B\,T^2\,h\,R_v\,d_h\big)$ in total (for all $t, u$).

Putting it all together, the total flops for multi-head attention with batch size $B$ are:

$$\underbrace{\mathcal{O}\big(B\,T^2\,R_q\,R_k\,d_h\big)}_{\substack{\text{QK b-dot products}\\ \text{(shared across heads)}}} + \underbrace{\mathcal{O}\big(B\,T^2\,h\,R_q\,R_k\big)}_{\text{per-head QK scalar mult.}} + \underbrace{\mathcal{O}\big(B\,T\,R_v\,d_h\big)}_{\substack{\text{Compute } \mathbf{b}_v \text{ for all tokens}\\ \text{(shared across heads)}}} + \underbrace{\mathcal{O}\big(B\,T^2\,h\,R_v\,d_h\big)}_{\substack{\text{final accumulations}\\ \text{(per head)}}}.$$

**Discussion.** By contrast, standard multi-head attention typically requires $\mathcal{O}\big(B\,T^2\,h\,d_h\big)$ flops for the $\mathbf{QK}^\top$ dot product (plus a similar $\mathcal{O}\big(B\,T^2\,h\,d_h\big)$ for multiplying by $\mathbf{V}$). The factorization can yield savings provided $R_q R_k \ll h$ (for QK) and $R_v \ll h$ (for V), though actual speedups depend on how well these multi-stage kernels are implemented and on hardware efficiency. By retaining $\mathbf{Q}, \mathbf{K}$, and $\mathbf{V}$ in factorized form, one can forgo the usual steps:

$$\mathbf{x}_t \mapsto \mathbf{Q}_t, \ \mathbf{K}_\tau \mapsto (\mathbf{Q}\,\mathbf{K}^\top) \mapsto \mathrm{softmax}(\mathbf{Q}\,\mathbf{K}^\top)\,\mathbf{V} \mapsto \text{final output}.$$

Instead, the large $\mathbf{Q}, \mathbf{K}, \mathbf{V}$ tensors (of size $T \times d_h$) are never materialized. The cost is replaced by rank-based b-dot-product computations plus per-head scalar multiplications. The main challenge is to keep the factor ranks $(R_q, R_k, R_v)$ sufficiently small relative to $d_h$ and to implement the necessary multi-stage kernels efficiently. When $R_q, R_k, R_v \ll h$, fully factorized QKV attention can yield substantial gains in both computation and memory footprint.

## A.8 Decoding Speed during Inference Time of MHA, MQA, GQA, MLA, and TPA

Suppose we are in an autoregressive setting, decoding the current token $\mathbf{x}_T$ given cached keys and values (KV) from all previous tokens $\mathbf{x}_1, \ldots, \mathbf{x}_{T-1}$. For each attention head $i \in \{1, \ldots, h\}$, we store $\mathbf{K}_i \in \mathbb{R}^{T \times d_h}, \mathbf{V}_i \in \mathbb{R}^{T \times d_h}$. Below, we compare the flops needed by MHA, MQA, GQA, MLA, and TPA to compute the next-token logits during inference.

**MHA, MQA, and GQA.** Despite sharing or grouping keys/values in MQA and GQA, the *decoding* cost for MHA, MQA, and GQA remains of the same order. Specifically, for each head $i$, we compute:

$$\mathbf{Q}_i(\mathbf{x}_T) \in \mathbb{R}^{d_h}, \quad \mathbf{K}_i \in \mathbb{R}^{T \times d_h}, \quad \mathbf{Q}_i(\mathbf{x}_T)\,\mathbf{K}_i^\top \in \mathbb{R}^{1 \times T}, \quad \text{and} \quad \mathrm{Softmax}\big(\mathbf{Q}_i(\mathbf{x}_T)\,\mathbf{K}_i^\top\big)\,\mathbf{V}_i \in \mathbb{R}^{d_h}.$$

Hence, the flops scale linearly in $h$, $d_h$, and $T$. For example, forming $\mathbf{Q}_i(\mathbf{x}_T)\mathbf{K}_i^\top$ for each head $i$ costs roughly $\mathcal{O}(h\,d_h\,T)$.

**MLA.** During inference, MLA can be seen as MQA but uses a larger head dimension to accommodate both RoPE and compressed representations (e.g., $d_h' = d_{\mathrm{rope}} + d_c$). In typical configurations, $d_{\mathrm{rope}} + d_c$ can be significantly larger (e.g., $d_h' = 576$ rather than $d_h = 64$ or $128$), thus inflating the dot-product cost by roughly $4.5\times$ to $9\times$ compared to MHA/MQA/GQA.

**TPA.** Recall that TPA factorizes $\mathbf{Q}$ and $\mathbf{K}$ into rank-$(R_q, R_k)$ terms (see Section A), potentially avoiding large $\mathbf{Q}, \mathbf{K}$ materializations. At inference, TPA's dot-product cost can be broken into two parts:

$$\underbrace{R_q\,R_k\,d_h\,T}_{\text{QK } \mathbf{b}\text{-dot products (shared across all heads)}} + 2 \underbrace{R_q\,R_k\,h\,T}_{\text{per-head scalar multiplications}},$$

where $T$ is the current sequence length. For concrete values $d_h = 128$, $h = 64$, $R_q = 8$, and $R_k = 2$ (or $R_q = 16$, $R_k = 1$), we obtain:

$$\text{MHA, MQA, GQA:} \quad 128 \times 64 \times T = 8192\,T,$$
$$\text{MLA:} \quad 576 \times 64 \times T = 36{,}384\,T,$$
$$\text{TPA:} \quad \big(8 \times 2 \times 128 \times T\big) + \big(2 \times 8 \times 2 \times 64 \times T\big) = 4096\,T.$$

Thus, in this setup, TPA can significantly reduce the flops needed for computing the $\mathbf{Q}(\mathbf{x}_T)\mathbf{K}^\top$ operation at each decoding step. The actual end-to-end wall-clock speedup also depends on kernel fusion, caching strategies, and hardware implementation details, but the factorized formulation offers a pathway to more efficient decoding than standard attention.

# B  Higher-Order Tensor Product Attention

All prior discussions have focused on a *second-order* factorization in which each rank-$R_Q$ (and similarly $R_K$, $R_V$) component is the outer product of two vectors: one in $\mathbb{R}^h$ (the "head" dimension) and one in $\mathbb{R}^{d_h}$. We now generalize this by introducing an additional latent factor, yielding a *third-order* (or higher) factorization reminiscent of canonical polyadic (CP) decomposition. Concretely, for a single token $t$, we write

$$\mathbf{Q}_t = \frac{1}{R_Q} \sum_{r=1}^{R_Q} \mathbf{a}_r^Q(\mathbf{x}_t) \otimes \mathrm{vec}\big(\mathbf{b}_r^Q(\mathbf{x}_t) \otimes \mathbf{c}_r^Q(\mathbf{x}_t)\big),$$

where the newly introduced factor $\mathbf{c}_r^Q(\mathbf{x}_t) \in \mathbb{R}^{d_c}$ can be viewed as a learnable gate or modulation term. Analogous expansions apply to $\mathbf{K}_t$ and $\mathbf{V}_t$. In practice, these triple (or higher-order) products still collapse into a matrix in $\mathbb{R}^{h \times d_h}$. One straightforward way to achieve this collapse is to split the feature dimension $d_h$ such that $d_b \times d_c = d_h$,

$$\mathbf{b}_r^Q(\mathbf{x}_t) \in \mathbb{R}^{d_b}, \quad \mathbf{c}_r^Q(\mathbf{x}_t) \in \mathbb{R}^{d_c}, \quad \mathrm{vec}\big(\mathbf{b}_r^Q(\mathbf{x}_t) \otimes \mathbf{c}_r^Q(\mathbf{x}_t)\big) \in \mathbb{R}^{d_h}.$$

This additional factor can enhance expressiveness without necessarily increasing the base rank. Conceptually, it can act as a learnable nonlinearity or gating mechanism. One could also tie or share $\mathbf{c}_r^Q$ across queries, keys, and values, to reduce parameter overhead.

A similar setup holds for keys (with rank $R_K$) and values (with rank $R_V$). Although this extra dimension adds to the parameter count, it can reduce the required rank to achieve a certain level of representational power.

From a memory perspective, higher-order TPA still leverages factorized KV caching: only the factors $\mathbf{a}(\mathbf{x}_t), \mathbf{b}(\mathbf{x}_t)$, and $\mathbf{c}(\mathbf{x}_t)$ for each past token are cached. As usual, a trade-off arises between model capacity and the overhead of memory and computing. Nonetheless, moving from a rank-$\big(R_Q, R_K, R_V\big)$ matrix factorization to a higher-order tensor decomposition can provide additional flexibility and increased capacity.

## B.1  RoPE Compatibility in Higher-Order TPA

Rotary positional embeddings (RoPE) remain compatible even under higher-order factorizations. In second-order TPA, RoPE can be treated as an invertible blockwise linear map acting on the last dimension of $\mathbf{Q}_t$ or $\mathbf{K}_t$. The same argument carries over when a third factor $\mathbf{c}_r^Q(\mathbf{x}_t)$ is present. Suppose RoPE acts on the $\mathbf{b}_r^Q(\mathbf{x}_t)$ portion (of dimension size $d_b$), we have the following theorem.

**Theorem 2** (RoPE Compatibility in Higher-Order TPA). Consider the higher-order (3-order) Tensor Product Attention (TPA) query factorization

$$\mathbf{Q}_t = \frac{1}{R_Q} \sum_{r=1}^{R_Q} \mathbf{a}_r^Q(\mathbf{x}_t) \otimes \mathrm{vec}\big(\mathbf{b}_r^Q(\mathbf{x}_t) \otimes \mathbf{c}_r^Q(\mathbf{x}_t)\big) \in \mathbb{R}^{h \times d_h},$$

where $\mathbf{a}_r^Q(\mathbf{x}_t) \in \mathbb{R}^h$, $\mathbf{b}_r^Q(\mathbf{x}_t) \in \mathbb{R}^{d_b}$, $\mathbf{c}_r^Q(\mathbf{x}_t) \in \mathbb{R}^{d_c}$, with $d_c = \frac{d_h}{d_b}$. Define the RoPE-transformed query as $\widetilde{\mathbf{Q}}_t = \mathrm{RoPE}_t(\mathbf{Q}_t) = \mathbf{Q}_t \mathbf{T}_t$, where

$$\mathbf{T}_t = \mathbf{R}_t \otimes \mathbf{I}_{d_c} = \begin{pmatrix} \mathbf{R}_t & \cdots & \mathbf{0} & \mathbf{0} \\ \mathbf{0} & \mathbf{R}_t & \cdots & \mathbf{0} \\ \vdots & \vdots & \ddots & \vdots \\ \mathbf{0} & \mathbf{0} & \cdots & \mathbf{R}_t \end{pmatrix} \in \mathbb{R}^{d_h \times d_h},$$

and $\mathbf{R}_t \in \mathbb{R}^{d_b \times d_b}$ ($d_b \in \mathbb{Z}_+$ is even) is a block-diagonal matrix composed of $2 \times 2$ rotation matrices:

$$\mathbf{R}_t = \begin{pmatrix} \cos(t\theta_1) & -\sin(t\theta_1) & & & & \\ \sin(t\theta_1) & \cos(t\theta_1) & & & & \\ & & \cos(t\theta_2) & -\sin(t\theta_2) & & \\ & & \sin(t\theta_2) & \cos(t\theta_2) & & \\ & & & & \ddots & \\ & & & & & \cos(t\theta_{d_b/2}) & -\sin(t\theta_{d_b/2}) \\ & & & & & \sin(t\theta_{d_b/2}) & \cos(t\theta_{d_b/2}) \end{pmatrix},$$

for $t \in \{1, \ldots, T\}$ and $j \in \{1, \ldots, d_b/2\}$.

This construction ensures that RoPE rotates only the coordinates corresponding to $\mathbf{b}_r^Q(\mathbf{x}_t)$ while leaving $\mathbf{c}_r^Q(\mathbf{x}_t)$ unchanged. Under these conditions, the RoPE-transformed query $\mathrm{RoPE}_t(\mathbf{Q}_t)$ admits a higher-order TPA factorization of the same rank $R_Q$. Specifically, we have

$$\frac{1}{R_Q} \sum_{r=1}^{R_Q} \mathbf{a}_r^Q(\mathbf{x}_t) \otimes \mathrm{vec}\left( \widetilde{\mathbf{b}}_r^Q(\mathbf{x}_t) \otimes \mathbf{c}_r^Q(\mathbf{x}_t) \right) = \mathrm{RoPE}_t(\mathbf{Q}_t), \tag{B.1}$$

where $\widetilde{\mathbf{b}}_r^Q(\mathbf{x}_t) = \mathbf{R}_t \mathbf{b}_r^Q(\mathbf{x}_t)$.

Please see Appendix C.2 for the proof. For fourth-order or higher, this result still holds.

# C Proofs of Theorems

## C.1 Proof of Theorem 1

*Proof.* Because RoPE is a linear orthogonal transform, we can write

$$\widetilde{\mathbf{Q}}_t = \mathbf{Q}_t \, \mathbf{T}_t = \frac{1}{R_Q} \left( \mathbf{A}_Q(\mathbf{x}_t)^\top \mathbf{B}_Q(\mathbf{x}_t) \right) \mathbf{T}_t = \frac{1}{R_Q} \mathbf{A}_Q(\mathbf{x}_t)^\top \left( \mathbf{B}_Q(\mathbf{x}_t) \, \mathbf{T}_t \right),$$

where $\mathbf{T}_t$ is the block-diagonal matrix encoding RoPE. This allows us to define

$$\widetilde{\mathbf{B}}_Q(\mathbf{x}_t) = \mathbf{B}_Q(\mathbf{x}_t) \, \mathbf{T}_t,$$

thereby obtaining

$$\mathrm{RoPE}(\mathbf{Q}_t) = \frac{1}{R_Q} \mathbf{A}_Q(\mathbf{x}_t)^\top \widetilde{\mathbf{B}}_Q(\mathbf{x}_t).$$

Similarly, for the key tensor $\mathbf{K}_s$, we have

$$\widetilde{\mathbf{K}}_s = \mathbf{K}_s \, \mathbf{T}_s = \frac{1}{R_K} \left( \mathbf{A}_K(\mathbf{x}_s)^\top \mathbf{B}_K(\mathbf{x}_s) \right) \mathbf{T}_s = \frac{1}{R_K} \mathbf{A}_K(\mathbf{x}_s)^\top \left( \mathbf{B}_K(\mathbf{x}_s) \, \mathbf{T}_s \right),$$

which defines

$$\widetilde{\mathbf{B}}_K(\mathbf{x}_s) = \mathbf{B}_K(\mathbf{x}_s) \, \mathbf{T}_s,$$

and thus

$$\text{RoPE}(\mathbf{K}_s) = \frac{1}{R_K} \mathbf{A}_K(\mathbf{x}_s)^\top \widetilde{\mathbf{B}}_K(\mathbf{x}_s).$$

Now, consider the product of the rotated queries and keys:

$$\widetilde{\mathbf{Q}}_t \widetilde{\mathbf{K}}_s^\top = \frac{1}{R_Q R_K} \left( \mathbf{A}_Q(\mathbf{x}_t)^\top \widetilde{\mathbf{B}}_Q(\mathbf{x}_t) \right) \left( \mathbf{A}_K(\mathbf{x}_s)^\top \widetilde{\mathbf{B}}_K(\mathbf{x}_s) \right)^\top$$

$$= \frac{1}{R_Q R_K} \mathbf{A}_Q(\mathbf{x}_t)^\top \widetilde{\mathbf{B}}_Q(\mathbf{x}_t) \widetilde{\mathbf{B}}_K(\mathbf{x}_s)^\top \mathbf{A}_K(\mathbf{x}_s),$$

Since $\mathbf{T}_t$ and $\mathbf{T}_s$ encode positional rotations, the product $\mathbf{T}_t \mathbf{T}_s^\top$ corresponds to a relative rotation $\mathbf{T}_{t-s}$. Therefore, we can express the above as

$$\widetilde{\mathbf{Q}}_t \widetilde{\mathbf{K}}_s^\top = \frac{1}{R_Q R_K} \mathbf{A}_Q(\mathbf{x}_t)^\top \left( \mathbf{B}_Q(\mathbf{x}_t) \mathbf{T}_t \mathbf{T}_s^\top \mathbf{B}_K(\mathbf{x}_s)^\top \right) \mathbf{A}_K(\mathbf{x}_s)$$

$$= \frac{1}{R_Q R_K} \mathbf{A}_Q(\mathbf{x}_t)^\top \left( \mathbf{B}_Q(\mathbf{x}_t) \mathbf{T}_{t-s} \mathbf{B}_K(\mathbf{x}_s)^\top \right) \mathbf{A}_K(\mathbf{x}_s)$$

$$= \frac{1}{R_Q R_K} \mathbf{A}_Q(\mathbf{x}_t)^\top \left( \mathbf{B}_Q(\mathbf{x}_t) \mathbf{T}_{t-s} \right) \left( \mathbf{B}_K(\mathbf{x}_s)^\top \mathbf{A}_K(\mathbf{x}_s) \right)$$

$$= \left( \frac{1}{R_Q} \mathbf{A}_Q(\mathbf{x}_t)^\top \mathbf{B}_Q(\mathbf{x}_t) \mathbf{T}_{t-s} \right) \left( \frac{1}{R_K} \mathbf{A}_K(\mathbf{x}_s)^\top \mathbf{B}_K(\mathbf{x}_s) \right)^\top,$$

This shows that

$$\text{RoPE}_{t-s}(\mathbf{Q}_t) \mathbf{K}_s^\top = \widetilde{\mathbf{Q}}_t \widetilde{\mathbf{K}}_s^\top,$$

Focusing on individual heads $i$, the above matrix equality implies:

$$\text{RoPE}_{t-s}(\mathbf{q}_{t,i})^\top \mathbf{k}_{s,i} = \widetilde{\mathbf{q}}_{t,i}^\top \widetilde{\mathbf{k}}_{s,i},$$

where

$$\widetilde{\mathbf{q}}_{t,i} = \text{RoPE}(\mathbf{q}_{t,i}) = \mathbf{T}_t \mathbf{q}_{t,i} \in \mathbb{R}^{d_h}, \quad \widetilde{\mathbf{k}}_{s,i} = \text{RoPE}(\mathbf{k}_{s,i}) = \mathbf{T}_s \mathbf{k}_{s,i} \in \mathbb{R}^{d_h}.$$

This equality confirms that the relative positional encoding between queries and keys is preserved under TPA's factorization and RoPE's rotation. Thus, TPA maintains compatibility with RoPE. This completes the proof of Theorem 1. $\qquad\square$

### C.2  Proof of Theorem 2

*Proof.* We begin by observing that each term $\mathbf{a}_r^Q(\mathbf{x}_t) \otimes \text{vec}\big(\mathbf{b}_r^Q(\mathbf{x}_t) \otimes \mathbf{c}_r^Q(\mathbf{x}_t)\big)$ is an element of $\mathbb{R}^h \otimes \mathbb{R}^{d_h}$. Here, $\mathbf{b}_r^Q(\mathbf{x}_t) \in \mathbb{R}^{d_b}$, $\mathbf{c}_r^Q(\mathbf{x}_t) \in \mathbb{R}^{d_c}$, with $d_c = \frac{d_h}{d_b}$. Consequently, the tensor product $\mathbf{b}_r^Q(\mathbf{x}_t) \otimes \mathbf{c}_r^Q(\mathbf{x}_t)$ forms a $d_b \times d_c$ matrix, and its vectorization lies in $\mathbb{R}^{d_b \cdot d_c} = \mathbb{R}^{d_h}$.

Applying the RoPE transformation to a single summand yields

$$\text{vec}\big(\mathbf{b}_r^Q(\mathbf{x}_t) \otimes \mathbf{c}_r^Q(\mathbf{x}_t)\big) \mapsto \mathbf{T}_t \text{vec}\big(\mathbf{b}_r^Q(\mathbf{x}_t) \otimes \mathbf{c}_r^Q(\mathbf{x}_t)\big).$$

Since $\mathbf{T}_t$ is defined as the Kronecker product $\mathbf{R}_t \otimes \mathbf{I}_{d_c}$, where $\mathbf{R}_t \in \mathbb{R}^{d_b \times d_b}$ and $\mathbf{I}_{d_c}$ is the identity matrix of size $d_c \times d_c$, it follows that

$$\mathbf{T}_t \text{vec}\big(\mathbf{b}_r^Q(\mathbf{x}_t) \otimes \mathbf{c}_r^Q(\mathbf{x}_t)\big) = \text{vec}\big(\mathbf{R}_t \mathbf{b}_r^Q(\mathbf{x}_t) \otimes \mathbf{c}_r^Q(\mathbf{x}_t)\big).$$

This is because the Kronecker product with an identity matrix effectively applies the rotation $\mathbf{R}_t$ to the $\mathbf{b}_r^Q(\mathbf{x}_t)$ component while leaving $\mathbf{c}_r^Q(\mathbf{x}_t)$ unchanged.

Therefore, the RoPE transformation of a single summand becomes

$$\text{RoPE}_t\Big(\mathbf{a}_r^Q(\mathbf{x}_t) \otimes \text{vec}\big(\mathbf{b}_r^Q(\mathbf{x}_t) \otimes \mathbf{c}_r^Q(\mathbf{x}_t)\big)\Big) = \mathbf{a}_r^Q(\mathbf{x}_t) \otimes \text{vec}\big(\mathbf{R}_t\mathbf{b}_r^Q(\mathbf{x}_t) \otimes \mathbf{c}_r^Q(\mathbf{x}_t)\big).$$

Importantly, this transformation does not mix the components $\mathbf{b}_r^Q(\mathbf{x}_t)$ and $\mathbf{c}_r^Q(\mathbf{x}_t)$; it solely rotates $\mathbf{b}_r^Q(\mathbf{x}_t)$ via $\mathbf{R}_t$. Summing over all ranks $r = 1, \ldots, R_Q$, we obtain

$$\frac{1}{R_Q} \sum_{r=1}^{R_Q} \mathbf{a}_r^Q(\mathbf{x}_t) \otimes \text{vec}\big(\mathbf{R}_t\mathbf{b}_r^Q(\mathbf{x}_t) \otimes \mathbf{c}_r^Q(\mathbf{x}_t)\big) = \text{RoPE}_t\big(\mathbf{Q}_t\big),$$

which retains the same higher-order TPA structure with rank $R_Q$.

Thus, the RoPE transformation is fully compatible with higher-order TPA, preserving the factorization rank and maintaining the structure by only rotating the $\mathbf{b}_r^Q(\mathbf{x}_t)$ components while leaving $\mathbf{c}_r^Q(\mathbf{x}_t)$ unchanged. ☐

# D  More Related Works

**Low-Rank Factorizations.** Low-rank approximations have been applied to compress model parameters and reduce complexity including LoRA (Hu et al., 2022), which factorizes weight updates during fine-tuning, and its derivatives for other training scenarios such as efficient pretraining (ReLoRA (Lialin et al., 2023), MoRA (Jiang et al., 2024)), long-context training (LongLoRA (Chen et al., 2024), SinkLoRA (Zhang, 2024)), as well as continual training (InfLoRA (Liang & Li, 2024), GS-LoRA (Zhao et al., 2024), I-LoRA (Ren et al., 2024)). These approaches typically produce static low-rank expansions that do not explicitly depend on the input context. And Malladi et al. (2023); Zeng & Lee (2024) provided theoretical proof of the expressiveness of low-rank approximation. For the initialization of factorization matrices, OLoRA (Büyükakyüz, 2024) applied QR-decomposition of pretrained weight to achieve better performance of language models while LoLDU (Shi et al., 2024) used LDU-decomposition to accelerate training of LoRA. Moreover, AdaLoRA (Zhang et al., 2023a) utilized Singular Value Decomposition (SVD) of the pretrained weight and introduced importance score for each parameter as a measurement to achieve dynamic adjustment of rank. TPA, by contrast, constructs Q, K, and V as contextually factorized tensors, enabling dynamic adaptation.

# E  More on Attention Mechanisms

## E.1  Multi-Query Attention (MQA)

Multi-Query Attention (MQA) (Shazeer, 2019) significantly reduces memory usage by *sharing* keys and values across heads, while still preserving unique query projections. For a sequence of embeddings $\mathbf{X} \in \mathbb{R}^{T \times d_{\text{model}}}$,

$$\mathbf{Q}_i = \mathbf{X}\boldsymbol{W}_i^Q, \quad \mathbf{K}_{\text{shared}} = \mathbf{X}\boldsymbol{W}_{\text{shared}}^K, \quad \mathbf{V}_{\text{shared}} = \mathbf{X}\boldsymbol{W}_{\text{shared}}^V.$$

Hence, each head $i$ only has a distinct query $\mathbf{Q}_i \in \mathbb{R}^{T \times d_h}$, but shares the same key $\mathbf{K}_{\text{shared}} \in \mathbb{R}^{T \times d_h}$ and value $\mathbf{V}_{\text{shared}} \in \mathbb{R}^{T \times d_h}$. In practice, this means:

$$\boldsymbol{W}_i^Q \in \mathbb{R}^{d_{\text{model}} \times d_h}, \quad \boldsymbol{W}_{\text{shared}}^K, \boldsymbol{W}_{\text{shared}}^V \in \mathbb{R}^{d_{\text{model}} \times d_h}.$$

The resulting MQA operation is:

$$\text{MQA}(\mathbf{X}) = \text{Concat}\Big(\textbf{head}_1, \ldots, \textbf{head}_h\Big)\boldsymbol{W}^O,$$

where

$$\textbf{head}_i = \text{Attention}\big(\mathbf{Q}_i, \mathbf{K}_{\text{shared}}, \mathbf{V}_{\text{shared}}\big).$$

By sharing these key and value projections, MQA cuts down on memory usage (especially for the key-value cache in autoregressive inference) but loses some expressivity since all heads must rely on the same key/value representations.

## E.2 Grouped Query Attention (GQA)

Grouped Query Attention (GQA) (Ainslie et al., 2023) generalizes MHA and MQA by *grouping* heads. Specifically, we partition the $h$ total heads into $G$ groups. Each group has a single set of keys and values, but each individual head within that group still retains its own query projection. Formally, if $g(i)$ maps a head $i \in [h]$ to its group index $g \in [G]$, then:

$$\mathbf{K}_{g(i)} = \mathbf{X}\,\boldsymbol{W}_{g(i)}^K, \quad \mathbf{V}_{g(i)} = \mathbf{X}\,\boldsymbol{W}_{g(i)}^V, \quad \mathbf{Q}_i = \mathbf{X}\,\boldsymbol{W}_i^Q,$$

and

$$\text{head}_i = \text{Attention}\Big(\mathbf{Q}_i, \mathbf{K}_{g(i)}, \mathbf{V}_{g(i)}\Big).$$

Again, $\boldsymbol{W}_g^K, \boldsymbol{W}_g^V \in \mathbb{R}^{d_{\text{model}} \times d_h}$ for each group $g$, and $\boldsymbol{W}_i^Q \in \mathbb{R}^{d_{\text{model}} \times d_h}$ for each head $i$. The complete output is again a concatenation of all heads:

$$\text{GQA}(\mathbf{X}) = \text{Concat}\Big(\text{head}_1, \ldots, \text{head}_h\Big)\boldsymbol{W}^O.$$

By adjusting $G$ between 1 and $h$, GQA can interpolate between sharing all key/value projections across heads (i.e., MQA) and having one set of projections per head (i.e., MHA).

## E.3 Multi-head Latent Attention (MLA)

Below, we briefly outline the Multi-head Latent Attention (MLA) approach used by DeepSeek-V2 (Liu et al., 2024a) and DeepSeek-V3 (Liu et al., 2024b). MLA introduces a low-rank compression of the keys and values to reduce the Key-Value (KV) caching cost at inference.

$$\mathbf{C}^{KV} = \mathbf{X}\boldsymbol{W}^{DKV},$$

$$\text{Concat}\big(\mathbf{K}_1^C, \mathbf{K}_2^C, \ldots, \mathbf{K}_h^C\big) = \mathbf{K}^C = \mathbf{C}^{KV}\boldsymbol{W}^{UK},$$

$$\mathbf{K}^R = \text{RoPE}\big(\mathbf{X}\boldsymbol{W}^{KR}\big),$$

$$\mathbf{K}_i = \text{Concat}\big(\mathbf{K}_i^C, \mathbf{K}^R\big),$$

$$\text{Concat}\big(\mathbf{V}_1^C, \mathbf{V}_2^C, \ldots, \mathbf{V}_h^C\big) = \mathbf{V}^C = \mathbf{C}^{KV}\boldsymbol{W}^{UV},$$

where $\boldsymbol{W}^{DKV} \in \mathbb{R}^{d_{\text{model}} \times d_c}, \boldsymbol{W}^{UK} \in \mathbb{R}^{d_c \times d_h h}, \boldsymbol{W}^{KR} \in \mathbb{R}^{d_{\text{model}} \times d_h^R}, \boldsymbol{W}^{UV} \in \mathbb{R}^{d_c \times d_h h}$, and $\mathbf{C}^{KV} \in \mathbb{R}^{T \times d_c}$ is the compressed KV latent (with $d_c \ll d_h h$), and $\text{RoPE}(\cdot)$ represents the RoPE transform applied to the separate key embeddings $\mathbf{K}^R$ of dimension $d_h^R$. Thus, only $\mathbf{C}^{KV}$ and $\mathbf{K}^R$ need to be cached, reducing KV memory usage while largely preserving performance compared to standard MHA (Vaswani et al., 2017).

MLA also compresses the queries, lowering their training-time memory footprint:

$$\mathbf{C}^Q = \mathbf{X}\boldsymbol{W}^{DQ},$$

$$\text{Concat}\big(\mathbf{Q}_1^C, \mathbf{Q}_2^C, \ldots, \mathbf{Q}_h^C\big) = \mathbf{Q}^C = \mathbf{C}^Q\boldsymbol{W}^{UQ},$$

$$\text{Concat}\big(\mathbf{Q}_1^R, \mathbf{Q}_2^R, \ldots, \mathbf{Q}_h^R\big) = \mathbf{Q}^R = \text{RoPE}\big(\mathbf{C}^Q\boldsymbol{W}^{QR}\big),$$

$$\mathbf{Q} = \text{Concat}\big(\mathbf{Q}^C, \mathbf{Q}^R\big).$$

where $\boldsymbol{W}^{DQ} \in \mathbb{R}^{d_{\text{model}} \times d_c'}, \boldsymbol{W}^{UQ} \in \mathbb{R}^{d_c' \times d_h h}, \boldsymbol{W}^{QR} \in \mathbb{R}^{d_c' \times d_h^R h}$. Here, $\mathbf{C}^Q \in \mathbb{R}^{T \times d_c'}$ (with $d_c' \ll d_h h$) is the compressed query latent. As above, each $\boldsymbol{W}^{DQ}, \boldsymbol{W}^{UQ},$ and $\boldsymbol{W}^{QR}$ connects these lower-dimensional query latents back to $h$ heads of dimension $d_h + d_h^R$.

Given compressed queries, keys, and values, the final attention output for the $t$-th token is:

$$\mathbf{O}_i = \text{Softmax}\Big(\frac{\mathbf{Q}_i\mathbf{K}_i^\top}{\sqrt{d_h + d_h^R}}\Big)\mathbf{V}_i^C,$$

$$\mathbf{U} = \text{Concat}\big(\mathbf{O}_1, \mathbf{O}_2, \ldots, \mathbf{O}_h\big)\boldsymbol{W}^O,$$

where $\boldsymbol{W}^O \in \mathbb{R}^{(d_h h) \times d_{\text{model}}}$ is the output projection.

In inference time, $\mathbf{C}^{KV}$ and $\mathbf{K}^R$ can be cached to accelerate decoding. In detail, when RoPE is ignored, the inner product $\mathbf{q}_{t,i}^\top \mathbf{k}_{s,i}$ (where $\mathbf{q}_{t,i}, \mathbf{k}_{s,i} \in \mathbb{R}^d$) of the $i$-th head between $t$-th and $s$-th tokens can be calculated using the hidden state $\mathbf{x}_t \in \mathbb{R}^{d_{\text{model}}}$ for $t$-th token and the cached latent state $\mathbf{c}_s^{KV} \in \mathbb{R}^{d_c}$ for $s$-th token:

$$\mathbf{q}_{t,i}^\top \mathbf{k}_{s,i} = [(\boldsymbol{W}_i^{UQ})^\top (\boldsymbol{W}_i^{DQ})^\top \mathbf{x}_t]^\top [(\boldsymbol{W}_i^{UK})^\top \mathbf{c}_s^{KV}] \tag{E.1}$$

$$= \mathbf{x}_t^\top [\boldsymbol{W}_i^{DQ} \boldsymbol{W}_i^{UQ} (\boldsymbol{W}_i^{UK})^\top] \mathbf{c}_s^{KV}, \tag{E.2}$$

where $\boldsymbol{W}_i^{(\cdot)}$ is the $i$-th head of the original weight, and $[\boldsymbol{W}_i^{DQ} \boldsymbol{W}_i^{UQ} (\boldsymbol{W}_i^{UK})^\top]$ can be computed previously for faster decoding. However, this process fails when RoPE is considered according to (Su, 2024). Since RoPE can be considered as multiplication with a block-diagonal matrix $\mathbf{T}_t \in \mathbb{R}^{d_h \times d_h}$ (see Section 2.3), with the property (2.1) that $\mathbf{T}_t \mathbf{T}_s^\top = \mathbf{T}_{t-s}$, then

$$\mathbf{q}_{t,i}^\top \mathbf{k}_{s,i} = [\mathbf{T}_t^\top (\boldsymbol{W}_i^{UQ})^\top (\boldsymbol{W}_i^{DQ})^\top \mathbf{x}_t]^\top [\mathbf{T}_s^\top (\boldsymbol{W}_i^{UK})^\top \mathbf{c}_s^{KV}]$$

$$= \mathbf{x}_t^\top [\boldsymbol{W}_i^{DQ} \boldsymbol{W}_i^{UQ} \mathbf{T}_{t-s} (\boldsymbol{W}_i^{UK})^\top] \mathbf{c}_s^{KV}. \tag{E.3}$$

Different from (E.2), acceleration by pre-computing $[\boldsymbol{W}_i^{DQ} \boldsymbol{W}_i^{UQ} \mathbf{T}_{t-s} (\boldsymbol{W}_i^{UK})^\top]$ fails since it varies for different $(t, s)$ position pairs. Therefore, MLA adds the additional $\mathbf{k}_t^R$ part with a relatively smaller size for RoPE compatibility. In Section 3.2, we will show that TPA addresses the issue of RoPE-incompatibility by applying tensor product.

$$\mathbf{C}^{KV} = \mathbf{X} \boldsymbol{W}^{DKV},$$

$$\text{Concat}(\mathbf{K}_1^C, \mathbf{K}_2^C, \ldots, \mathbf{K}_h^C) = \mathbf{K}^C = \mathbf{C}^{KV} \boldsymbol{W}^{UK},$$

$$\mathbf{K}^R = \text{RoPE}(\mathbf{X} \boldsymbol{W}^{KR}),$$

$$\mathbf{K}_i = \text{Concat}(\mathbf{K}_i^C, \mathbf{K}^R),$$

$$\text{Concat}(\mathbf{V}_1^C, \mathbf{V}_2^C, \ldots, \mathbf{V}_h^C) = \mathbf{V}^C = \mathbf{C}^{KV} \boldsymbol{W}^{UV},$$

### E.4 Multi-matrix Factorization Attention (MFA)

Hu et al. (2024) proposed Multi-matrix Factorization Attention (MFA), which can be seen as Multi-Query Attention (MQA) with dimension of each head equals $d_C$, and low-rank factorized Q:

$$\mathbf{Q}_i = \mathbf{X} \boldsymbol{W}^{DQ} \boldsymbol{W}_i^{UQ}, \quad \mathbf{K}_{\text{shared}} = \mathbf{X} \boldsymbol{W}_{\text{shared}}^K, \quad \mathbf{V}_{\text{shared}} = \mathbf{X} \boldsymbol{W}_{\text{shared}}^V,$$

where

$$\boldsymbol{W}^{DQ} \in \mathbb{R}^{d_{\text{model}} \times d_c}, \quad \boldsymbol{W}_i^{UQ} \in \mathbb{R}^{d_c \times d_c}, \quad \boldsymbol{W}_{\text{shared}}^K, \boldsymbol{W}_{\text{shared}}^V \in \mathbb{R}^{d_{\text{model}} \times d_c}.$$

## F Other Variants of TPA

**TPA with Non-contextual B.** Conversely, one may fix the token-dimension factors $\mathbf{b}_r^Q, \mathbf{b}_r^K, \mathbf{b}_r^V \in \mathbb{R}^{d_h}$ as learned parameters, while allowing $\mathbf{a}_r^Q(\mathbf{x}_t), \mathbf{a}_r^K(\mathbf{x}_t), \mathbf{a}_r^V(\mathbf{x}_t)$ to adapt to $\mathbf{x}_t$. For keys:

$$\mathbf{K}_t = \frac{1}{R_K} \sum_{r=1}^{R_K} \mathbf{a}_r^K(\mathbf{x}_t) \otimes \mathbf{b}_r^K,$$

and similarly for values. This arrangement is effective if the token-dimension structure remains mostly uniform across the sequence, while the head-dimension factors capture context.

**TPA KV Only.** One can preserve a standard query mapping,

$$\mathbf{Q}_t = \boldsymbol{W}^Q \mathbf{x}_t \in \mathbb{R}^{h \times d_h},$$

and factorize only the keys and values. This leaves the query projection as the original linear transformation while reducing memory usage via factorized KV caching.

**TPA KV with Shared B.** Another variant is to share the token-dimension factors of keys and values:

$$\mathbf{b}_r^K(\mathbf{x}_t) = \mathbf{b}_r^V(\mathbf{x}_t),$$

lowering parameter counts and the KV cache footprint. While it constrains $\mathbf{K}$ and $\mathbf{V}$ to be formed from the same token basis, it can still perform well and provide additional memory savings.

**Nonlinear Head Factors.** Rather than applying purely linear mappings to the head-dimension factors $\mathbf{a}_r^Q, \mathbf{a}_r^K, \mathbf{a}_r^V$, one may introduce element-wise nonlinearities such as $\sigma(\cdot)$ or $\text{softmax}(\cdot)$. This effectively yields a *Mixture of Heads Attention* (MoH Attention), where each component becomes a learned mixture weight modulated by the nonlinearity.

**Discussion.** These variants illustrate TPA's versatility in balancing memory cost, computational overhead, and representation power. By choosing which dimensions (heads or tokens) remain contextual and adjusting ranks $(R_Q, R_K, R_V)$, TPA unifies multiple existing attention mechanisms—such as MHA, MQA, and GQA—under one framework, while potentially reducing the KV cache size by an order of magnitude during autoregressive inference.

# G  More on Experiments

## G.1  Experimental Settings

We list the main architecture hyper-parameters and training devices in Table 4. We fix $d_h = 64$ for all the models. Moreover, we fix the number of KV heads with 2 for GQA models; $d_h^R = 32$ for MLA models; and $R_k = R_v = 2$, $R_q = 6$ for TPA and TPA-KV only models. Other hyper-parameters are listed in Table 5.

*Table 4.* The architecture hyper-parameters and training devices of models. Abbreviations: BS. = Batch Size, GAS. = Gradient Accumulation Steps.

| MODEL SIZE | #PARAM | DEVICES | MICRO BS. | GAS. | #LAYER | $d_{\text{MODEL}}$ |
|---|---|---|---|---|---|---|
| SMALL | 124M | 4× A100 GPUs | 24 | 5 | 12 | 768 |
| MEDIUM | 353M | 8× A100 GPUs | 20 | 3 | 24 | 1024 |
| LARGE | 772M | 8× A100 GPUs | 15 | 4 | 36 | 1280 |
| XL | 1.55B | 8× A100 GPUs | 6 | 10 | 48 | 1600 |

*Table 5.* The architecture hyper-parameters for different models.

| MODEL SIZE | SMALL | MEDIUM | LARGE | XL |
|---|---|---|---|---|
| $h$ (MHA) | 12 | 16 | 20 | 25 |
| $h$ (MQA) | 23 | 31 | 39 | 49 |
| $h$ (GQA) | 22 | 30 | 38 | 48 |
| $h$ (MLA) | 12 | 23 | 34 | 49 |
| $h$ (TPA-KVONLY) | 22 | 29 | 37 | 47 |
| $h$ (TPA) | 34 | 47 | 61 | 78 |
| $d_c$ (MLA) | 256 | 512 | 512 | 512 |
| $d_c'$ (MLA) | 512 | 1024 | 1024 | 1024 |

## G.2  Additional Experimental Results

### G.2.1  PERPLEXITY CURVES

We display the perplexity curves for medium, large and XL size of models in Figure 4.

### G.2.2  ABLATION STUDY ON DIFFERENT RANKS

Figure 5 shows the training loss, validation loss, and validation perplexity curves of XL-size (1.5B) T6 models with different ranks trained on the FineWeb-Edu 100B dataset, and the evaluation results are displayed in Table 7. It can be observed that increase in rank can improve the performances of large language models.

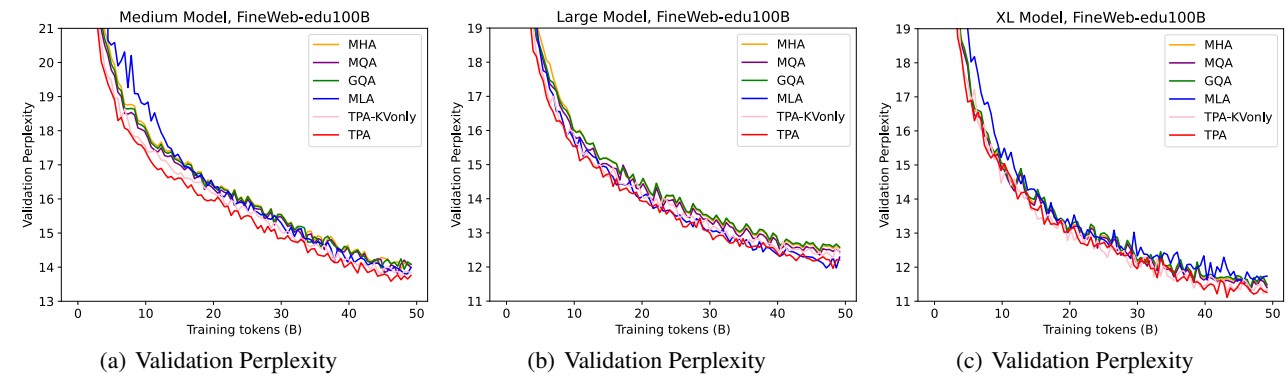

*Figure 4.* The validation perplexity of medium-size (353M) models, large-size (773M), and XL-size (1.5B) models with different attention mechanisms on the FineWeb-Edu 100B dataset.

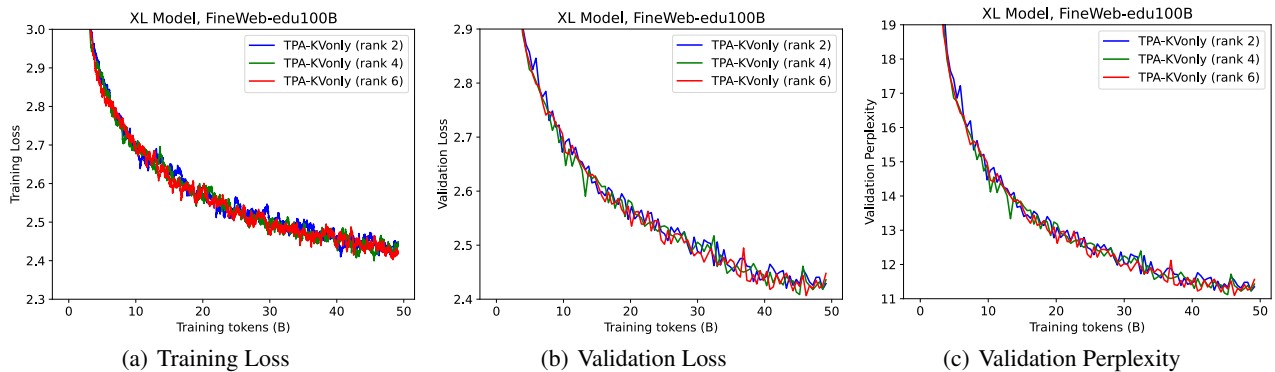

*Figure 5.* The training loss, validation loss and validation perplexity curves of XL-size (1.5B) T6 models with different ranks on the FineWeb-Edu 100B dataset.

### G.2.3 0-SHOT EVALUATION WITH LM-EVALUATION-HARNESS

For the evaluation, We show the 0-shot performances with lm-evaluation-harness for small-size (124M) and XL-size (1.5B) models in Tables 6 and 7.

### G.2.4 2-SHOT EVALUATION WITH LM-EVALUATION-HARNESS

We also show 2-shot performances in Tables 8, 9, 10 and 11.

### G.3 Ablation Studies on Learning Rates

We implement a set of parallel experiments for medium models with learning rate $3 \times 10^{-4}$, and the curves for training loss, validation loss, and validation perplexity are displayed in Figure 6. We also show the performance of these models on the benchmarks described in Section 4 in Tables 12-13. The results show that TPA and TPA-KVonly models can also outperform other types of attention with different learning rates.

*Table 6.* The evaluation results of small models with different attention mechanisms pre-trained using FineWeb-Edu 100B dataset (0-shot with lm-evaluation-harness). The best scores in each column are **bolded**. Abbreviations: HellaSw. = HellaSwag, W.G. = WinoGrande.

| Method | ARC-E | ARC-C | BoolQ | HellaSw. | OBQA | PIQA | W.G. | MMLU | SciQ | Avg. |
|---|---|---|---|---|---|---|---|---|---|---|
| MHA | 50.63 | 26.96 | **59.39** | 36.18 | 32.00 | 64.96 | **51.85** | 23.40 | 70.30 | 46.19 |
| MQA | 49.62 | 25.34 | 55.72 | 35.94 | 31.40 | 64.85 | 51.30 | 23.37 | 68.70 | 45.14 |
| GQA | 48.70 | 25.68 | 56.15 | 35.58 | 31.40 | 64.91 | 51.62 | 23.12 | 68.20 | 45.04 |
| MLA | 50.21 | 26.71 | 58.01 | 36.25 | **32.80** | 64.69 | 50.59 | 24.67 | 71.90 | 46.20 |
| **TPA-KVonly** | 51.05 | 26.54 | 57.25 | **36.77** | 32.60 | **65.02** | 50.91 | 23.64 | 69.70 | 45.94 |
| **TPA (non-ctx-A)** | 50.17 | 25.60 | 57.95 | 36.13 | 31.40 | 64.80 | 49.57 | **24.88** | 64.80 | 45.03 |
| **TPA** | **51.26** | **27.39** | 57.00 | 36.68 | **32.80** | 64.47 | 49.72 | 24.61 | **72.00** | **46.21** |

*Table 7.* The evaluation results of XL models with different attention mechanisms pre-trained using the FineWeb-Edu 100B dataset (0-shot with lm-evaluation-harness). The best scores in each column are **bolded**. Abbreviations: HellaSw. = HellaSwag, W.G. = WinoGrande. If not specified, TPA and TPA-KVonly set $R_K = R_V = 2$.

| Method | ARC-E | ARC-C | BoolQ | HellaSw. | OBQA | PIQA | W.G. | MMLU | SciQ | Avg. |
|---|---|---|---|---|---|---|---|---|---|---|
| MHA | 64.81 | 35.41 | 61.90 | 54.32 | 37.20 | 72.74 | 55.80 | 25.44 | **82.80** | 54.49 |
| MQA | 64.10 | 36.01 | 62.26 | 54.38 | 39.00 | 72.58 | 56.43 | 23.70 | 81.90 | 54.48 |
| GQA | 63.68 | 35.92 | 60.46 | 54.17 | 38.40 | **73.56** | 56.27 | 24.77 | 81.70 | 54.33 |
| MLA | 64.14 | 35.92 | 60.12 | 53.60 | 39.20 | 72.25 | 55.17 | 24.71 | 81.60 | 54.08 |
| **TPA-KVonly** | 65.61 | 36.77 | **63.02** | 54.17 | 37.00 | 73.34 | 54.62 | 25.02 | 81.60 | 54.57 |
| **TPA-KVonly ($R_{K,V} = 4$)** | 64.52 | **37.03** | 63.27 | **54.89** | 39.80 | 72.91 | 56.51 | 24.74 | 81.60 | **55.03** |
| **TPA-KVonly ($R_{K,V} = 6$)** | 65.78 | 35.92 | 61.71 | 54.86 | 38.60 | 72.69 | **57.93** | **25.59** | 82.20 | **55.03** |
| **TPA** | **66.71** | 36.52 | 61.38 | 54.03 | **40.40** | 72.52 | 56.83 | 24.49 | 82.20 | 55.01 |

*Table 8.* The evaluation results of small models with different attention mechanisms on FineWeb-Edu 100B dataset (2-shot with lm-evaluation-harness). The best scores in each column are **bolded**. Abbreviations: HellaSw. = HellaSwag, W.G. = WinoGrande.

| Method | ARC-E | ARC-C | BoolQ | HellaSw. | OBQA | PIQA | W.G. | MMLU | SciQ | Avg. |
|---|---|---|---|---|---|---|---|---|---|---|
| MHA | **57.66** | **28.24** | 57.28 | 36.43 | 29.60 | 64.09 | 51.14 | **26.57** | **82.00** | **48.11** |
| MQA | 53.79 | 26.35 | 44.95 | 34.18 | 28.80 | 62.79 | 52.01 | 25.91 | 78.10 | 45.21 |
| GQA | 55.01 | 25.94 | 55.72 | 35.68 | **31.80** | 65.29 | 51.93 | 25.27 | 77.80 | 47.16 |
| MLA | 54.76 | 27.13 | **58.07** | 36.13 | 31.40 | 65.07 | 51.30 | 25.90 | 78.90 | 47.63 |
| **TPA-KVonly** | 54.25 | 27.90 | 57.06 | 36.36 | **31.80** | 64.31 | **53.59** | 26.18 | 79.20 | 47.85 |
| **TPA (non-ctx-A)** | 55.09 | 27.65 | 53.82 | 36.24 | 30.20 | 64.53 | 50.75 | 26.01 | 78.60 | 46.99 |
| **TPA** | 57.53 | 28.07 | 56.33 | **36.49** | **31.80** | 64.36 | 51.14 | 25.92 | 79.70 | 47.93 |

*Table 9.* The evaluation results of medium models with different attention mechanisms pre-trained using FineWeb-Edu 100B dataset (2-shot with lm-evaluation-harness). The best scores in each column are **bolded**. Abbreviations: HellaSw. = HellaSwag, W.G. = WinoGrande.

| Method | ARC-E | ARC-C | BoolQ | HellaSw. | OBQA | PIQA | W.G. | MMLU | SciQ | Avg. |
|---|---|---|---|---|---|---|---|---|---|---|
| MHA | 64.73 | 32.42 | 58.29 | 45.89 | 34.20 | 68.50 | 53.20 | **25.86** | 88.00 | 52.34 |
| MQA | 64.98 | 33.62 | 55.02 | 45.81 | 34.00 | 69.59 | 53.43 | 24.30 | 85.20 | 51.77 |
| GQA | 65.24 | 33.19 | 56.54 | 45.41 | 34.80 | 69.04 | **55.72** | 24.73 | 87.90 | 52.51 |
| MLA | 64.98 | 33.62 | 53.52 | 45.94 | 33.00 | 68.55 | 51.85 | 25.46 | 89.10 | 51.78 |
| **TPA-KVonly** | 64.69 | 32.34 | **59.48** | 46.23 | **35.40** | **70.08** | 54.06 | 25.64 | 86.30 | 52.69 |
| **TPA (non-ctx-A)** | 65.45 | 33.79 | 56.88 | 45.23 | 33.60 | 68.61 | 54.22 | 25.00 | 85.00 | 51.98 |
| **TPA** | **67.97** | **34.56** | 57.22 | **46.87** | 34.60 | 69.91 | 52.01 | 25.07 | **89.90** | **53.12** |

*Table 10.* The evaluation results of large models with different attention mechanisms pre-trained using the FineWeb-Edu 100B dataset (2-shot with lm-evaluation-harness). The best scores in each column are **bolded**. Abbreviations: HellaSwag = HellaSwag, WG = WinoGrande.

| Method | ARC-E | ARC-C | BoolQ | HellaSwag | OBQA | PIQA | WG | MMLU | SciQ | Avg. |
|---|---|---|---|---|---|---|---|---|---|---|
| MHA | 67.85 | 36.35 | 59.82 | 50.22 | 35.00 | 70.67 | 53.35 | 23.92 | 91.10 | 54.25 |
| MQA | 68.86 | 36.09 | 53.79 | 50.50 | **37.00** | 70.89 | **54.70** | 25.01 | 88.00 | 53.87 |
| GQA | 69.15 | 36.09 | 58.84 | 50.29 | 36.20 | 70.73 | 54.22 | **26.08** | 90.00 | 54.62 |
| MLA | 70.54 | **38.74** | **61.50** | **51.86** | 36.00 | 70.89 | 54.22 | 25.47 | **92.40** | **55.74** |
| **TPA-KVonly** | **71.34** | 37.71 | 59.76 | 51.10 | 36.00 | **71.49** | 54.62 | 25.83 | 90.10 | 55.33 |
| **TPA** | 70.41 | 37.71 | 60.06 | 51.30 | 34.00 | 71.06 | 54.54 | 25.79 | 90.30 | 55.02 |

*Table 11.* The evaluation results of XL models with different attention mechanisms pre-trained using the FineWeb-Edu 100B dataset (2-shot with lm-evaluation-harness). The best scores in each column are **bolded**. Abbreviations: HellaSwag = HellaSwag, WG = WinoGrande. If not specified, We set $R_K = R_V = 2$ for TPA and TPA-KVonly.

| Method | ARC-E | ARC-C | BoolQ | HellaSwag | OBQA | PIQA | WG | MMLU | SciQ | Avg. |
|---|---|---|---|---|---|---|---|---|---|---|
| MHA | 70.83 | 39.93 | 59.85 | 54.05 | 36.20 | 72.52 | 55.17 | 25.42 | 91.70 | 56.18 |
| MQA | 71.34 | 39.76 | 58.93 | 54.27 | 39.40 | 72.96 | 57.38 | 24.74 | 91.90 | 56.74 |
| GQA | 71.17 | 39.08 | 60.18 | 54.05 | 37.40 | 73.07 | 56.35 | 24.87 | **92.20** | 56.49 |
| MLA | 70.79 | 37.54 | 50.83 | 53.33 | **40.00** | 72.09 | 56.51 | 24.93 | 91.80 | 55.31 |
| **TPA-KVonly** | 72.85 | 39.68 | 60.92 | 53.81 | 37.00 | **73.34** | 56.83 | **26.19** | 91.30 | 56.88 |
| **TPA-KVonly** ($R_{K,V} = 4$) | 72.98 | **40.27** | 60.15 | **54.88** | 36.80 | 73.29 | 56.43 | 25.50 | 92.10 | 56.93 |
| **TPA-KVonly** ($R_{K,V} = 6$) | **73.95** | 39.76 | 58.99 | 54.73 | 36.80 | 72.91 | **59.04** | 24.93 | 92.90 | **57.11** |
| **TPA** | 71.76 | 39.16 | **61.25** | 53.74 | 37.80 | 72.80 | 55.49 | 23.86 | 90.70 | 56.28 |

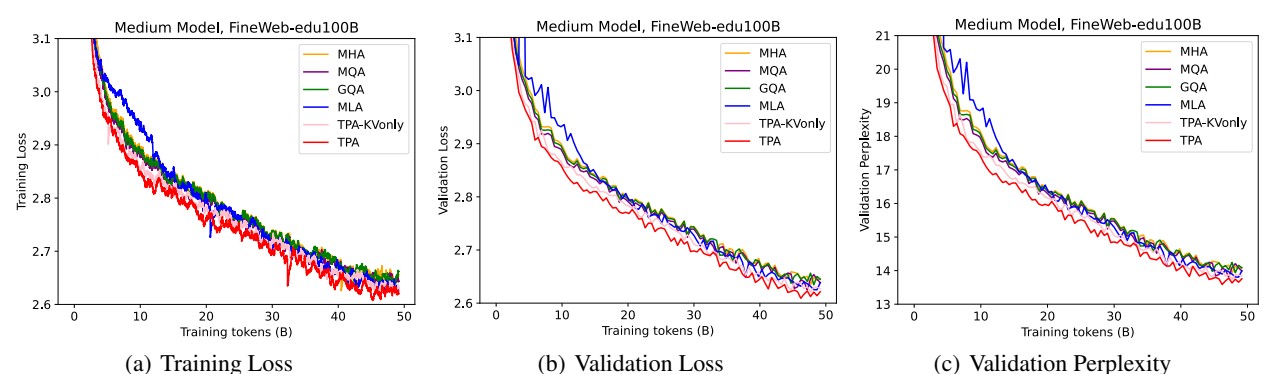

(a) Training Loss  (b) Validation Loss  (c) Validation Perplexity

*Figure 6.* The training loss, validation loss, and validation perplexity of medium-size (353M) models (learning rate $3 \times 10^{-4}$) and different attention mechanisms on the FineWeb-Edu 100B dataset.

*Table 12.* The evaluation results of medium models (learning rate $3 \times 10^{-4}$) with different attention mechanisms pretrained using the FineWeb-Edu 100B dataset (0-shot with lm-evaluation-harness). The best scores in each column are **bolded**. Abbreviations: HellaSw. = HellaSwag, W.G. = WinoGrande.

| Method | ARC-E | ARC-C | BoolQ | HellaSw. | OBQA | PIQA | W.G. | MMLU | SciQ | Avg. |
|---|---|---|---|---|---|---|---|---|---|---|
| MHA | 56.52 | 29.27 | 58.84 | 44.06 | 35.00 | 68.44 | 51.07 | 25.35 | 76.40 | 49.44 |
| MQA | 55.68 | 28.24 | 60.86 | 44.17 | **35.20** | 68.66 | 52.72 | 25.14 | 72.90 | 49.29 |
| GQA | 54.88 | 29.61 | 56.36 | 43.77 | **35.20** | 68.82 | 52.57 | **25.41** | 74.80 | 49.05 |
| MLA | **59.64** | 29.78 | 60.73 | 45.17 | 34.20 | 68.66 | 52.80 | 25.34 | 75.70 | 50.22 |
| **TPA-KVonly** | 57.11 | 30.03 | **61.25** | 44.83 | 34.60 | 69.04 | **54.54** | 23.35 | 74.60 | 49.93 |
| **TPA** | 59.30 | **31.91** | 60.98 | **45.57** | 34.60 | **69.48** | 53.91 | 24.93 | **77.20** | **50.88** |

*Table 13.* The evaluation results of medium models (learning rate $3 \times 10^{-4}$) with different attention mechanisms pre-trained using the FineWeb-Edu 100B dataset (2-shot with lm-evaluation-harness). The best scores in each column are **bolded**. Abbreviations: HellaSw. = HellaSwag, W.G. = WinoGrande.

| Method | ARC-E | ARC-C | BoolQ | HellaSw. | OBQA | PIQA | W.G. | MMLU | SciQ | Avg. |
|---|---|---|---|---|---|---|---|---|---|---|
| MHA | 64.44 | 32.85 | **59.05** | 44.18 | 33.20 | 68.72 | 50.12 | **26.01** | 87.40 | 49.44 |
| MQA | 64.27 | 32.94 | 57.71 | 44.36 | 31.80 | 68.01 | 51.70 | 25.99 | 86.00 | 49.29 |
| GQA | 61.70 | 32.17 | 52.81 | 43.99 | 33.80 | 68.50 | 53.35 | 24.44 | 86.40 | 50.80 |
| MLA | 65.95 | 31.48 | 50.98 | 44.99 | 32.20 | 68.93 | 51.93 | 25.89 | 88.80 | 51.24 |
| **TPA-KVonly** | 65.99 | 33.70 | 57.49 | 44.47 | **34.20** | **69.53** | 53.28 | 24.23 | 86.50 | 49.93 |
| **TPA** | **66.54** | **34.47** | 58.96 | **45.35** | 33.00 | 69.21 | **53.99** | 24.51 | **91.30** | **53.04** |

