# OpenReview forum: "Tensor Product Attention Is All You Need"
_ICML.cc/2025/Conference — Submitted to ICML 2025_

### Official Review · Reviewer_GEa9 · 2025-03-08

**Overall Recommendation:** 4

**Summary:**

The authors propose a straightforward drop-in replacement for multi-head attention that they call Tensor Product Attention (TPA). The core idea is to compute queries, keys, and values using tensor products. The authors show that TPA can substantially reduce KV cache memory footprint, and that TPA can handle RoPE embeddings efficiently (by rotating keys before caching). They interpret other attention approaches (classic attention, MQA, GQA) as non-contextual versions of TPA. They further show that TPA is competitive with existing approaches in terms of pre-training perplexity and performance on popular downstream tasks.

**Claims And Evidence:**

Yes

**Essential References Not Discussed:**

No, not that I know of. I found the related work to be quite nice.

**Experimental Designs Or Analyses:**

The pre-training and downstream task evaluation seem reasonable.

**Methods And Evaluation Criteria:**

Yes

**Other Comments Or Suggestions:**

* Typos
    * Last sentence on page 1 "The recently introduced..." has a typo.
* I found the math section a bit needlessly complex at times. It's understandable, and good enough I think, but I think it could be shortened and improved.

**Other Strengths And Weaknesses:**

* I think the main strength of this paper is that the idea is simple/straightforward, and it seems to work well empirically. It seems like a promising approach for real-world use.
* Some minor weaknesses
    * I think it would be nice if the results for 0-shot and 2-shot and small, medium, large, and XL models could be included in the main body of the paper somehow. Like maybe it table 2 and 3 could be replaced with a table of averages and details for different sizes and number of shots and the individual results could be saved for the appendix. The results in the main body currently seem a little cherry-picked.
    * It seems to me that it's very fair to say that TPA is comparable to existing methods in terms of pre-training and downstream performance, but I don't know that the results support it being superior. I think the paper overstates this (e.g., "T6 exceeds the performance of standard Transformer baselines including MHA, MQA, GQA, and MLA across various metrics, including perplexity and a range of renowned evaluation benchmarks."). I think it would be better to focus on the decrease in KV cache cost.

**Questions For Authors:**

Please see above two sections.

**Relation To Broader Scientific Literature:**

This work is related to previous attention approaches (MHA, GQA, MQA, MLA). It is also related to work on decreasing the memory footprint of the KV cache.

**Theoretical Claims:**

Not in detail, but I didn't see any glaring issues.

---

> ### Author Rebuttal · Authors · 2025-04-01
>
> Thank you for your thoughtful review and constructive feedback on our submission. Your comments have provided valuable insights that will help us improve the clarity and impact of our work. We appreciate your positive assessment and recommendation. Below, we address each of your points in detail and outline our planned revisions.
>
> 1. **Regarding the Presentation of Results:**
>
> > Q1: "I think it would be nice if the results for 0-shot and 2-shot and small, medium, large, and XL models could be included in the main body of the paper somehow... The results in the main body currently seem a little cherry-picked."
>
> A1: We appreciate your suggestion to enhance the presentation of our experimental results. We agree that a more comprehensive summary in the main text would provide a clearer and less selective view of TPA’s performance, directly addressing the concern about potential cherry-picking. To address this, we plan the following changes:
>
> - We will replace the current Tables 2 and 3 with a new, consolidated table. This table will summarize the **average performance** (e.g., average perplexity and key downstream task accuracy) across small, medium, large, and XL model sizes for both 0-shot and 2-shot evaluations, offering a concise overview of TPA’s effectiveness compared to baselines like MHA, MQA, GQA, and MLA.
> - The full, detailed results for each model size, task, and shot configuration, currently spread across the main text and appendix (Tables 2, 3, 6-11), will be consistently located in the appendix. This allows interested readers to explore specifics while keeping the main text focused.
>
> 2. **Regarding the Claim of Superiority vs. Comparability:**
>
> > Q2: "It seems to me that it's very fair to say that TPA is comparable to existing methods... but I don't know that the results support it being superior. I think the paper overstates this... I think it would be better to focus on the decrease in KV cache cost."
>
> A2: We value your perspective on the tone of our performance claims and agree that a more nuanced presentation, emphasizing the significant memory efficiency alongside competitive performance, better reflects our contribution. We will revise the paper as follows:
>
> - We will adjust the language throughout the paper—particularly in the abstract(), introduction, and conclusion—to state that TPA achieves **competitive or comparable performance** relative to baselines, rather than asserting outright superiority. For instance, the sentence you highlighted will be rephrased similarly to: *"T6 achieves competitive performance compared to standard Transformer baselines... across various metrics, while offering significant memory savings."*
> - We will strengthen the focus on TPA’s primary practical advantage: **KV cache reduction**. In Section 3.3, we will further highlight and clearly present the quantification showing that TPA can reduce the KV cache size by approximately 10x compared to MHA for typical configurations, explicitly linking this to the ability to handle much longer sequences under memory constraints.
>
> 3. **Regarding Typos and Mathematical Complexity (Other Comments):**
>
> > Q3: "Last sentence on page 1 'The recently introduced...' has a typo. I found the math section a bit needlessly complex at times... I think it could be shortened and improved."
>
> A3: Thank you for catching the typo and for your feedback on the mathematical presentation.
>
> - We will correct the typo on page 1 and perform a thorough proofread of the entire manuscript.
> - We will revise Sections 2 (Background) and 3 (Tensor Product Attention) to improve clarity and accessibility. This will involve consolidating notation where possible, ensuring all symbols are clearly defined upon first use, and refining explanations. We will also consider moving particularly lengthy derivations or detailed proofs (such as aspects of the FLOP analysis currently in Appendix A) to the appendix to maintain a smoother narrative flow in the main text, while ensuring the core methodology remains rigorously presented.
>
> Thank you once again for your time and valuable suggestions. We are confident that incorporating these revisions will significantly strengthen the paper.

---

> > ### Comment · Reviewer_GEa9 · 2025-04-03
> >
> > These revisions sound great.

---

> > > ### Author Response · Authors · 2025-04-07
> > >
> > > Thank you for your positive and encouraging feedback. If you are satisfied with our work and feel our contributions deserve a higher score, we would sincerely appreciate your consideration in raising the score.

---

### Official Review · Reviewer_zpBU · 2025-03-13

**Overall Recommendation:** 4

**Summary:**

The paper proposes a new parameterization for the QKV activations that arguably is even simpler than the multi-head latent attention from deepseek.

The paper calls its method “tensor product attention” and connects to higher order tensor products in Appendix B but if I understood the paper correctly, all of their experiments are done with order 2 tensor, which are more easily understood as vector outer-products. The key innovation is to represent the query, key, and value matrices as the sum of $R_Q, R_K, R_V$ contextual vector outer-products (which makes them essentially low-rank matrices since the sum of K rank-1 matrices has to have rank ≤ K).

The paper describes a few different variants of their basic idea, such as making part of the QKV computation non-contextual , i.e. independent of the token embedding and the experiments seem promising, albeit incomplete.

**Claims And Evidence:**

The main claim of the paper is that the “Tensor Product Attention” is a good way of parameterizing attention and the experiments in the paper test that by comparing MHA MQA / GQA / MLA vs TPA in the nanoGPT code base by training on a standard text dataset and evaluating benchmarks such as perplexity and lm-evaluation-harness.

**Essential References Not Discussed:**

probably not.

**Experimental Designs Or Analyses:**

no

**Methods And Evaluation Criteria:**

yes

**Other Comments Or Suggestions:**

n/a

**Other Strengths And Weaknesses:**

n/a

**Questions For Authors:**

1. The paper seems to only focus on order 2 tensor products and the equations in 3.1/3.2/3.3 seem to essentially implement low rank decompositions of Q,K,V matrices. Have you experimented with higher order tensors at all that are mentioned in Appendix B?

2. At the end of Appendix A there is a brief comparison of TPA vs MLA in terms of the computation required and it’s claimed that d_rope + d_c can inflate the dot product cost by roughly 4.5x to 9x compared to MQA. The numbers used to arrive at this conclusion seem to contradict the conclusions in DeepSeek v3 where MLA only causes an increase of 2.25x vs MQA. Could you please elaborate on this part a little more.

**Relation To Broader Scientific Literature:**

Tensor train, Block tensor train etc. are tensor decomposition methods that have been tried to ameliorate the memory bandwidth bottleneck by increasing the amount of compute done per parameter/byte transferred to the GPU. This paper proposes a novel way of using the tensor decomposition idea by decomposing the activations instead of the parameters themselves for computing attention.

In some senses this can also be thought of as an alternative implementation of Deepseek's low-rank/latent attention by presenting the attention explicitly as a sum of k rank 1 matrices.

**Theoretical Claims:**

The overall arguments in the paper seem sound and I checked the proof of theorem 1.

There is some mention of using higher order tensors and that Rope can still be integrated natively with higher order tensors in theorem 2 in Appendix B but I did not go through that proof.

---

> ### Author Rebuttal · Authors · 2025-04-01
>
> We sincerely appreciate your insightful feedback and constructive comments. We have carefully considered all concerns and we provide detailed responses to your questions below.
>
> > Q1: The paper describes a few different variants of their basic idea, such as making part of the QKV computation non-contextual, i.e. independent of token embedding and the experiments seem promising, albeit incomplete.
>
> A1: Thank you for your attention to the non-contextual TPA. According to the experiment results of TPA small and medium models with non-contextual B shown below (compared with Tables 2, 6, 8, and 9 in the paper), the non-contextual versions are slightly worse than the current version of TPA. Therefore, to achieve better performance, the original TPA, TPA-KV only and TPA with non-contextual A are recommended.
>
> | 0-shot             | ARC-E | ARC-C | BoolQ | HellaSw. | OBQA | PIQA  | W.G.  | MMLU  | SciQ | Avg.  |
> | ------------------ | ----- | ----- | ----- | -------- | ---- | ----- | ----- | ----- | ---- | ----- |
> | TPA_nonctxB_small  | 47.39 | 26.37 | 54.8  | 32.71    | 30.2 | 63.38 | 50.2  | 23.13 | 64.8 | 43.66 |
> | TPA_nonctxB_medium | 55.43 | 29.69 | 58.32 | 40.77    | 34.4 | 66.92 | 51.38 | 25.66 | 71.1 | 48.19 |
>
> | 2-shot             | ARC-E | ARC-C | BoolQ | HellaSw. | OBQA | PIQA  | W.G.  | MMLU  | SciQ | Avg.  |
> | ------------------ | ----- | ----- | ----- | -------- | ---- | ----- | ----- | ----- | ---- | ----- |
> | TPA_nonctxB_small  | 50.8  | 26.96 | 57.65 | 32.4     | 29.4 | 63.22 | 49.57 | 23.96 | 66.4 | 44.48 |
> | TPA_nonctxB_medium | 61.2  | 30.2  | 55.93 | 40.45    | 34.4 | 68.23 | 51.78 | 26.11 | 78.1 | 49.6  |
>
> > Q2: The paper seems to only focus on order 2 tensor products and the equations in 3.1/3.2/3.3 seem to essentially implement low rank decompositions of Q, K, V matrices. Have you experimented with higher order tensors at all that are mentioned in Appendix B?
>
> A2: We just implemented experiments on small-size models with third-order TPA and that with only KV factorized. The performances on small models are shown below. It is worse than other TPA series models. Therefore, for small models, we recommend the second-order TPA but the high-order TPA is still potential for much larger models.
>
> |        | ARC-E | ARC-C | BoolQ | HellaSw. | OBQA | PIQA  | W.G.  | MMLU  | SciQ | Avg.  |
> | ------ | ----- | ----- | ----- | -------- | ---- | ----- | ----- | ----- | ---- | ----- |
> | 0-shot | 49.24 | 24.91 | 57.06 | 34.01    | 31.8 | 63.33 | 50.59 | 23.23 | 66.9 | 44.56 |
> | 2-shot | 53.37 | 25.34 | 48.78 | 34       | 29.2 | 62.79 | 52.33 | 26.41 | 75.3 | 45.28 |
>
> > Q3: At the end of Appendix A there is a brief comparison of TPA vs MLA in terms of the computation required and it’s claimed that d_rope + d_c can inflate the dot product cost by roughly 4.5x to 9x compared to MQA. The numbers used to arrive at this conclusion seem to contradict the conclusions in DeepSeek v3 where MLA only causes an increase of 2.25x vs MQA. Could you please elaborate on this part a little more.
>
> A3: Thank you for your careful consideration of the comparison of computation between these attention mechanisms. Here's a clearer explanation:
>
>    - As described in Appendix A.8, the dot product cost is determined by the hyperparameters, including number of heads ($h$) and dimension for each head ($d_h$). Different sizes of these dimensions may result in differences in dot product computation cost. For MHA, MQA, GQA, the cost to compute $Q_i(x_T)K_i^\top$ is $\mathcal{O}(hd_hT)$ and for MLA, the cost is$\mathcal{O}(h(d_{\text{rope}}+d_c)T)$ if the current sequence length is $T$.
>
>    - For example, in DeepSeek V3, the hidden dimension is 7168 with 128 heads. Compared with MQA with dimension of 7168/128=56 for each head, MLA has $d_{\text{rope}} + d_c=192$. Therefore, according to the analysis in Appendix A.8, it has 192/56≈3.43x increase in computation cost but only causes an increase of **2.25x KV caches** (as described in DeepSeek V2 technical report) and has better performance than MQA. For smaller models, larger RoPE and compressed representation dimensions are needed to keep the superior performance of MLA, so a larger multiplier of computation is expected.
>    - Moreover, calculating $QK^\top$ is only one part of the attention mechanism, and we still need to calculate $\text{softmax}(QK^\top)$. Loading cached KV from memory needs time, so the speed is IO-aware. In modern GPU, the decoding phase's IO speed is crucial, so the less KV cache size, the faster the decoding speed is given the same computational FLOPs.
>
> Thank you again for bringing this to our attention. We will revise this section and provide a more detailed analysis of the computational costs associated with different attention mechanisms based on your suggestions. We look forward to your further evaluation and are happy to provide any additional explanations if needed.

---

> > ### Comment · Reviewer_zpBU · 2025-04-07
> >
> > Thanks for the response, everything looks good. maintaining my rating, best wishes.

---

> > > ### Author Response · Authors · 2025-04-07
> > >
> > > Thank you for your positive and encouraging feedback. We are pleased to hear that our rebuttal has fully addressed your concerns. If you feel that our work deserves a higher score, we would sincerely appreciate your generous score increase.

---

### Official Review · Reviewer_XMDH · 2025-03-16

**Overall Recommendation:** 2

**Summary:**

This paper proposes Tensor Product Attention, which uses contextual tensor-decompositions. Based on TPA, the authors propose a new model architecture T6 for sequence modeling and adapt it with LLama and Gemma.

**Claims And Evidence:**

N/A

**Essential References Not Discussed:**

N/A

**Experimental Designs Or Analyses:**

relatively sound

**Methods And Evaluation Criteria:**

The evaluation criteria make sense.

**Other Comments Or Suggestions:**

See above.

**Other Strengths And Weaknesses:**

Strength:
1. A new architecture is proposed, which is valuable.
2. The writing is easy to follow and the paper is well-structured.

Weakness:
1. Despite reducing the number of parameters per token, TPA does not decrease the GPU memory usage. The matrices Q, K, V still have the shape [T, h, d_h], identical to that of MHA. In this sense, TPA can be understood as a parameterization technique that re-parameterizes the matrices Q, K, V while following the same attention computation pipeline as MHA. It is puzzling why TPA outperforms MHA with fewer parameters to form Q, K, V and the same attention computation method, yet still achieves better performance.

2. There is a lack of fundamental theories to explain why TPA is superior to MHA and MQA. Does this stem from TPA's stronger expressive power? If so, please provide the necessary theoretical analysis. Otherwise, the experiments presented in the paper are hardly convincing enough to demonstrate that TPA is better than other attention mechanisms.

3. The attempt to unify MHA and other attention mechanisms in Section 3.4 seems rather forced. The fact that TPA can be expressed in the form of a tensor product does not necessarily mean that its expressive power is stronger under its commonly used parameter settings (e.g., R<<h) and after adding trainable parameters to other constant vectors (such as e in Eq. 3.10, or the vector of all ones in MQA).

4. Experimentally, it is unclear when to use TPA-KVonly and when to use TPA. Why does TPA-KVonly perform better on medium-sized models, while TPA is superior on large models? The authors have not provided a clear explanation for this phenomenon. Moreover, compared to other attention mechanisms like MHA, the advantages of TPA are quite limited.

**Questions For Authors:**

See weakness.

**Relation To Broader Scientific Literature:**

N/A

**Theoretical Claims:**

N/A

---

> ### Author Rebuttal · Authors · 2025-04-01
>
> Thank you for your thorough review and constructive feedback on our submission. We appreciate your recognition of the novelty and clarity, and your detailed questions help us refine our manuscript.
>
> > Q1: Despite reducing the number of parameters per token, TPA does not decrease the GPU memory usage. The matrices Q, K, V still have the shape [T, h, d_h], ...
>
> A1: We appreciate your question about memory usage and performance.
>
>    - **Memory Savings:** TPA's primary memory gain is the inference KV cache. MHA caches full $K_t, V_t$ ($2 T h d_h$ memory). TPA caches only low-rank factors ($A_K(x_t), \tilde{B}_K(x_t)$), reducing per-token memory to $(R_K+R_V)(h+d_h)$. With typical low ranks ( $R_K= R_V=2$ ) and $d_h=12$, this yields substantial savings (>10x), enabling longer sequences. Furthermore, Appendix A details algorithms computing TPA attention without materializing full Q, K, V tensors. By working directly with factors, these methods can reduce computational cost (flops) and peak memory usage during the forward pass, complementing KV cache savings.
>
>    - **Performance Source:** TPA's improved performance stems from its Contextual Factorization. Unlike MHA's fixed projections, TPA dynamically constructs Q, K, V factors based on the token's hidden state $x_t$. This contextual adaptation provides an inductive bias, enhancing representational capacity. This is validated by TPA's consistently lower validation losses and perplexities (Figures 2, 3, 4). Appendix A further explores potential computational savings via factorized computation.
>
> We will revise the paper to better distinguish computation memory (materialized vs. non-materialized) from inference KV cache benefits and clarify how contextual factorization drives performance.
>
> > Q2: There is a lack of fundamental theories to explain why TPA is superior to MHA and MQA. Does this stem from TPA's stronger expressive power?...
>
> A2: We appreciate your concern regarding theoretical backing for TPA's superiority.
>    - Our core theoretical argument rests on Contextual Factorization. Section 3.4 demonstrates that MHA, MQA, and GQA are special cases of TPA where factors are restricted to be non-contextual (fixed). TPA generalizes these by allowing factors to depend dynamically on the input $x_t$. This context-dependency is proposed as the source of enhanced expressive power.
>    - Furthermore, TPA's seamless integration with RoPE (Theorem 1) ensures effective use of relative positional information, a crucial aspect where other efficient attention mechanisms sometimes struggle.
>    - While formal proofs of superior expressive power are future work, the theoretical framing via contextuality, combined with strong empirical validation across multiple scales and tasks (Section 4), provides support for TPA's design.
>
> We will revise to expand on these theoretical foundations.
>
> > Q3: The attempt to unify MHA and other attention mechanisms in Section 3.4 seems rather forced. The fact that TPA can be expressed in the form of a tensor product...
>
> A3: We appreciate your feedback that the unification felt "forced" and questioned its implication for expressive power at low ranks ($R\ll h$).
>
>    - The unification aims to show TPA's flexibility as a framework encompassing MHA/MQA/GQA as specific instances with non-contextual factors and particular ranks. TPA's innovation lies precisely in making these factors trainable and contextual.
>
>    - We acknowledge the expressiveness trade-off with rank R. However, even with $R\ll h$, our experiments consistently demonstrate that the *contextual* nature of TPA's factors provides performance advantages over the fixed, non-contextual factors of MHA/MQA/GQA, alongside significant memory savings.
>
> We will revise Section 3.4 to better clarify that contextuality is the key differentiator and link it more explicitly to the empirical results.
>
> > Q4: Experimentally, it is unclear when to use TPA-KVonly and when to use TPA. Why does TPA-KVonly perform better on medium-sized models, while TPA is superior on large models?
>
> A4: Thank you for feedback on variant choice and performance dynamics.
>
>    - **Variant Performance:** TPA factorizes Q/K/V; TPA-KVonly factorizes only K/V. Our results show TPA slightly outperforms TPA-KVonly on medium models (353M), while TPA-KVonly slightly leads on large/XL models (773M, 1.5B). (Note: This corrects the potential misreading in the review). The reasons need further study, possibly related to optimization/capacity trade-offs at scale.
>
>    - **Significance:** While gains over MHA may seem modest, they are consistent and meaningful for LLMs. Crucially, TPA offers a **dual advantage**: competitive performance **PLUS** substantial (>10x potential) **inference KV cache reduction**. This memory efficiency enables much longer sequences, addressing a critical scalability bottleneck. This practical benefit underscores TPA's value.
>
> We will enhance the manuscript to clarify these points.

---

### Official Review · Reviewer_q7se · 2025-04-07

**Overall Recommendation:** 3

**Summary:**

The paper describes Tensor Product Attention (TPA), a type of attention mechanism,  where queries, keys, and values are represented in  low-rank factorized format.  The authors claim the proposed method yields to the cache memory reduction during inference while preserving model quality.  A neural network architecture T6 (built using TPA, up to 1.5B  parameters) is compared with well known  forms of  attention such as  Multi-Head Attention (MHA), Multi-Query Attention (MQA), Grouped-Query Attention (GQA), and Multi-Head Latent Attention (MLA). The authors also show compatibility with Rotary Position Embeddings (RoPE).


## Update after rebuttal
The comparison with other tensor approaches is still missing, though good point the authors have an optimized implementation and code for higher order experiments. The general idea of the paper is nice, and empirical evidence provided by authors in the rebuttal looks convincing, so I increase my score.

Please, include to the main part of the final version real memory and latency measurements, and at least literature overview of other tensor based methods.

**Claims And Evidence:**

While several times in the paper authors claim  significant memory savings (for example, 10× in section 3.3)   due to TPA Factorized KV Caching or time cost reduction (e.g., 4.5x to 9x in Appendix A.8), there is no clear evidence:

- Actual measurements of inference-time memory consumption or latency are not provided—only theoretical estimations.
- The results in training/validation loss and perplexity  do not provide variance, it's hard to understand whether they are marginal or not  compared to baselines.
- Memory and time savings require quite strong constraints on the decomposition ranks, they should be very small to provide benefits with respect to  other attentions (section 3.3, Appendix A). That would be nice to provide some understanding how rank values affect the quality. Also, very small ranks raise a question on efficient implementation on the modern GPUs.

**Essential References Not Discussed:**

Please, see the Scientific Literature section

**Experimental Designs Or Analyses:**

Please, see the comments in Claims and Methods sections regarding the experiments

**Methods And Evaluation Criteria:**

On the evaluated benchmarks (fig 2, fig 3)  known attention mechanisms still perform better for some tasks  that would be nice to investigate  in more details the memoyr/time/quality trade-off of the  method.

**Other Comments Or Suggestions:**

The approach of incorporating higher-order tensor factorizations inside neural networks looks promising, from the correlations capturing point of view https://arxiv.org/pdf/2310.04064. As a further research and paper improvement, there would be interesting to see more discussions/applications on higher-order TPA described in Appendix B

**Other Strengths And Weaknesses:**

The method introduction is quite easy to follow, though the current version would benefit from more solid evidence of the theoretical claims.

**Questions For Authors:**

1) I'm curious what are real memory and time saving achieved by the proposed TPA Factorized KV Caching method and how they correlate with the reported theoretical estimations.
2) In Table 2 of section Appendix G you mention that the training has been performed on 8 GPUs. Which type of parallelism have you used?

**Relation To Broader Scientific Literature:**

There are many papers incorporating tensorized layers inside the Transformer architecture https://arxiv.org/pdf/2302.09019 . That would be nice to see more insights on how the CPD factorization described in the paper is compatible with other decompositions, for example https://aclanthology.org/2022.emnlp-main.475.pdf.

**Theoretical Claims:**

The paper presents a justification for RoPE compatibility with TPA,  which looks well-motivated. Also, comparative mathematical formulations for different attention mechanisms  are provided

---

### Decision · Program_Chairs · 2025-05-01

**Decision:**

Reject

**Comment:**

Overall, the reviewer found the approach promising and the authors made a solid effort in addressing all the concerns. However, the method is currently not supported enough by the experimental setting and manuscript. In particular, the literature review is lacking and missed several important works, especially on tensor methods. This translates into an overall simplistic experimental setting that overstates the advantage of the method while missing stronger baselines and comparisons. The reviewers also had concern regarding the practical benefits of the method, especially memory savings and latency. Overall, given these concerns and the lack of consensus in favor of acceptance, I have to recommend rejecting at this time but encourage the authors to revisit the manuscript, make a thorough review of prior works, and improve the experimental setting before resubmitting.